

# A user perspective on the avalanche danger scale – Insights from North America

Abby Morgan[1], Pascal Haegeli[1], Henry Finn[1, 2], Patrick Mair[3]

[1]School of Resource and Environmental Management, Simon Fraser University, Burnaby, V5A 1S6, Canada
[2]School of Social and Political Science, University of Edinburgh, Edinburgh, EH8 9LD, UK
[3]Department of Psychology, Harvard University, Cambridge, MA 02138, United States

*Correspondence to*: Pascal Haegeli (pascal_haegeli@sfu.ca)

**Abstract.** Danger ratings are used across many fields to convey the severity of a hazard. In snow avalanche risk management,
danger ratings play a prominent role in public bulletins by concisely describing existing and expected conditions. While there
is considerable research examining the accuracy and consistency of the production of avalanche danger ratings, far less
research has focused on how backcountry recreationists interpret and apply the scale.

We used 3195 responses to an online survey to provide insight into how recreationists perceive the North American Public
Avalanche Danger Scale and how they use ratings to make trip planning decisions. Using a latent class mixed effect model,
our analysis shows that the most common perception of the danger scale is linear. People with a linear perception expect the
hazard to increase in a stepwise fashion between levels. This understanding is contrary to the scientific understanding of the
scale, which indicates an exponential-like increase in severity between levels. Regardless of perception, most respondents
report avoiding the backcountry at the two highest ratings. Using conditional inference trees, we show that participants who
recreate fewer days per year and those who have lower levels of avalanche safety training tend to rely more heavily on the
danger rating to make trip planning decisions. These results provide avalanche warning services with a better understanding
of how recreationists interact with danger ratings and highlight how critical the ratings are for individuals who recreate less
often and who have lower levels of training. We discuss opportunities for avalanche warning services to optimize the danger
scale to meet the needs of these users who depend on the ratings the most.

## 1 Introduction

A central goal of risk communication is the provision of accurate, timely, and trustworthy information that empowers people
to make informed decisions about mitigative actions (Lundgren and McMakin, 2018; National Oceanic and Atmospheric
Administration, 2016). In the field of natural and environmental hazards, danger or hazard scales are a common method to
communicate the severity of current or expected conditions to the public in a simple way. We see examples of these scales in
contexts such as forest fire danger ratings (Province of British Columbia, 2022), air quality health indexes (Environment and





Climate Change Canada, 2022), heat and humidity measures (National Oceanic and Atmospheric Administration, 2022), and

many more.

Snow avalanches are another context where a danger scale plays a prominent role in risk communication. Snow avalanches are a serious natural hazard in mountainous regions around the world that can threaten settlements, transportation corridors, critical infrastructure (e.g., transmission lines), natural resource extraction (e.g., timber harvesting and mining), as well as

people and infrastructure at remote worksites. In addition, recreationists pursuing backcountry activities, such as backcountry skiing, mountain snowmobiling, ice climbing, or snowshoeing, voluntarily expose themselves to avalanche hazard in many countries.

Public avalanche forecasts (also known as bulletins) published by local avalanche warning services are a critical source of information for recreationists during trip planning. To make the avalanche hazard information accessible to backcountry users

with different levels of avalanche training and education, bulletins communicate avalanche conditions in a tiered format that presents information in layers with increasing levels of detail and complexity. This approach, which is commonly referred to as the information pyramid (European Avalanche Warning Services, 2021b), is designed to maximize comprehension across audiences with varying avalanche education, experience and needs. The first information that recreationists see when they consult the public avalanche bulletins is the avalanche danger rating, which communicates the general severity of avalanche

conditions in a region over a certain amount of time (Statham et al., 2010). The scales that have been used to communicate avalanche hazard have evolved over time, ranging from four to eight levels across iterations of the danger scale in both North America and Europe (Dennis and Moore, 1996; Mitterer and Mitterer, 2018). However, an ordinal five-level scale with standardized colours, signal words, numbers, and icons has been used most consistently by public avalanche warning services around the world to describe the conditions. The current version of the North American Public Avalanche Danger Scale

(Statham et al., 2010; Figure 1) is closely tied to the Conceptual Model of Avalanche Hazard (Statham et al., 2018a), which defines the key elements of avalanche hazard and provides a workflow for consistent avalanche hazard assessments in North America. While there are slight variations in the signal words and level definitions between the danger scales used in Europe and North America, the main difference is that the primary focus of the North American Public Avalanche Danger Scale is public risk communication, whereas the European system is used in a wider range of applications that also includes providing

warnings for residential areas and transportation networks (Stoffel & Meister, 2004; Stoffel & Schweizer, 2008).





**Figure 1: Current representation of the North American Public Avalanche Danger Scale**

Despite being used as a critical public risk communication tool since 1994, the North American Public Avalanche Danger
Scale has not had a comprehensive analysis of how the target audience uses it. Instead, much of the existing research on the
danger scale has focused on the production of danger ratings. Consistently producing accurate and credible avalanche hazard
assessments is challenging not only due to variability and uncertainty in the data informing the forecast, but also because the
human judgment involved in the assessment process is susceptible to interpretation and bias (Statham et al., 2018b). Several
recent studies have focused on identifying sources of bias or error and improving the production of accurate and credible
danger ratings (Clark, 2019; Lazar et al., 2016; Schweizer et al., 2020, 2021; Techel and Schweizer, 2017).

In contrast to the research efforts focused on improving the quality and consistency of avalanche bulletins, there has been
relatively little research focused on how recreationists are perceiving and using the forecast products, including the danger
scale. Best practices in risk communication stress that to communicate the severity of conditions effectively, risk
communicators must not only provide accurate and credible risk information from a trusted source, but also interact with the
target audience to understand their knowledge and perspectives. This information is critical for crafting appropriate messages
that resonate with the audience (Lundgren and McMakin, 2018; National Research Council, 1996; National Oceanic and
Atmospheric Administration, 2016). Applying these principles to an avalanche context, Winkler and Techel (2014) examined
recreationists' assessment of the quality of the bulletin website design compared to previous renditions. Engeset et al. (2018)
examined the effectiveness of Norwegian avalanche risk communication products and suggested that although bulletin users
considered danger ratings an important piece of information, the ratings alone were not enough information to communicate
intended warning information. St. Clair et al. (2019) developed a typology that describes the different ways recreationists use
avalanche bulletin information, and Finn (2020) investigated bulletin literacy amongst different recreation user types and
provided targeted risk communication recommendations for specific user groups. One of the few studies that explicitly
examined how recreationists use the North American Public Avalanche Danger Scale was conducted by Ipsos Reid (2009) in
support of its last revision in 2010. Using an online survey, the study examined the effect of the scale's revised definition on



recreationists' ability to identify appropriate terrain. While recreationists presented with the new descriptors alone made more conservative terrain choices under Considerable than with the definitions of the old scale, they reverted to their original terrain choices when the descriptors were presented together with the signal words (Ipsos Reid, 2009).

The lack of research on the perception and use of danger scales does not seem limited to the avalanche safety community. A
literature review on research on danger scales in general revealed that natural hazard risk communication literature tends to focus on how the public interacts with warnings, alerts, and orders in an emergency or crisis context. Examples include how people responded to mandatory hurricane evacuation orders (Demuth et al., 2018), flood risk and communication perception (Kellens et al., 2013), flash flood mental models and misunderstandings (Lazrus et al., 2016), tornado warnings (Brotzge and Donner, 2013), and pre-crisis earthquake risk communication (Herovic et al., 2020). While these topics are related to danger
ratings, the voluntary nature of interacting with natural hazards in recreational settings (e.g., backcountry travel in avalanche terrain, ocean activities with wave hazards and rip currents, and recreating in canyons prone to flash floods) creates a unique context where the transferability of the existing research results may be limited. As far as we are aware, there has been no user-focused research on danger scales in the context of voluntary hazards to date.

Given the importance of having an in-depth understanding of the risk message audience, the avalanche safety community has
a considerable knowledge gap in understanding how recreationists are interacting with the avalanche danger rating, which may limit the effectiveness of this risk communication tool. The purpose of this research is to contribute to a better understanding of the strength and weaknesses of the avalanche danger scale by exploring how recreationists perceive and use it during their trip planning process to decide whether to go into the backcountry.

## 2 Methods

### 2.1 Survey Design

To gain insight into how winter backcountry recreationists in Canada and the United States use, understand, and apply the safety information presented in daily avalanche bulletins published by local warning services, the Simon Fraser University Avalanche Research Program conducted a large online survey in the spring of 2019. The overall objective of this survey was to provide avalanche warning services with empirical evidence for making decisions about how to improve their avalanche
risk communication products. While the survey included a wide range of exercises and questions, only the design of the main questions of interest for the analysis are presented in this paper. Readers interested in the design of the entire survey are referred to Finn (2020) for a more complete description.

To classify participants based on the bulletin use typology described by St. Clair et al. (2021), the survey presented them with an ordered series of statements (Table 1) and asked them to indicate which of these statements describes their personal bulletin
use practice most accurately. The available options included an additional statement for a Bulletin User Type F that was not part of the original typology. This addition allowed recreationists who do not use the avalanche bulletin for trip planning to be





properly separated from avalanche safety professionals who do not use the avalanche bulletin because they have access to professional information sources.

After indicating their avalanche bulletin use practices, Bulletin User Type As were split from the rest of the sample as most

survey questions were only relevant for participants familiar with avalanche bulletins. To gain some insight about the intuitiveness of the avalanche danger scale for recreationists with limited familiarity, the survey presented Type A participants with the five signal words (Low, Moderate, Considerable, High, and Extreme) in a random order and asked them to arrange the words in order of severity.

For all other bulletin user types, the survey included several questions targeting participants' understanding and use of the

danger scale. To provide detailed insight about where recreationists might be challenged with the danger scale, the questions were designed using Krathwohl's (2002) adaptation of Bloom's taxonomy of learning objectives (Bloom, 1956), which is an education framework that identifies increasingly complex learning processes. Reflecting the first three stages of the learning process described by Krathwohl (2002), our survey questions were designed to shed light on participants' recall, understanding, and use of danger ratings.


**Table 1: Statements included in avalanche bulletin user type question**

| Bulletin User Type | Statements |
|---|---|
| Type A | It is not typical for me to consult avalanche bulletin information when making my backcountry travel plans. |
| Type B | I typically use the bulletin to check the danger rating, which informs my decision of whether or not it's safe to travel in the backcountry. |
| Type C | I typically combine the danger rating from the bulletin with knowledge of how avalanche prone an area is to determine where to travel in the backcountry. |
| Type D | I typically make a decision about where or when to go based on the specific nature of the avalanche problem conditions reported in the bulletin and whether I feel that I can manage my travel in the terrain given these conditions. |
| Type E | I typically use the available information about the specific nature of the avalanche problem conditions from the bulletin as a starting point for my continuous assessment in the field to confirm or disconfirm the information where I am travelling. |
| Type F | It is not typical for me to consult public avalanche bulletins or forecasts because I have access to professional information sources (e.g., InfoEx) that offer more detailed insight into current conditions. |

To better understand how well participants knew the danger rating terminology, the danger rating section of our survey started with a question that prompted participants to recall the danger rating levels in their proper order. To answer this question,

participants had to type the levels from least to most severe into open-text field.





To examine participants' understanding of the danger scale we designed a question that examined how their perception of the scale aligns with how avalanche researchers view the scale. The question included sliders where participants could indicate how they perceive the severity of each level of the danger scale on a numeric scale from 0 (no avalanche hazard at all) to 100 (widespread, large natural avalanches reaching valley bottoms) (Figure 2). All sliders moved in increments of 2, and the design

of the question did not restrict the slider movement at all, which means that participants were able to have overlapping severity ranges or gaps between them.

a) Starting position

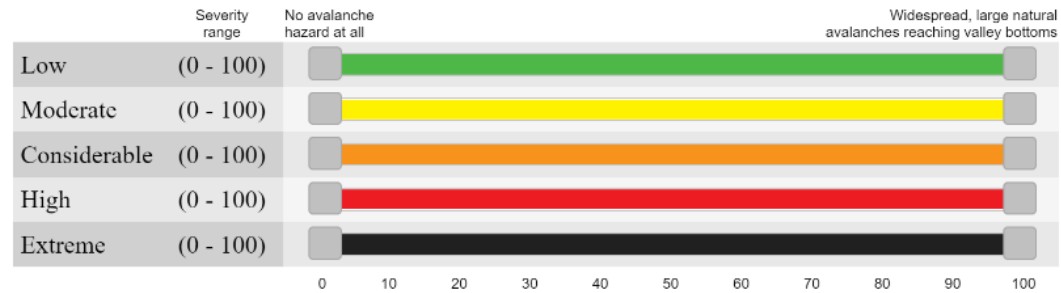

140                                                    b) Example answers

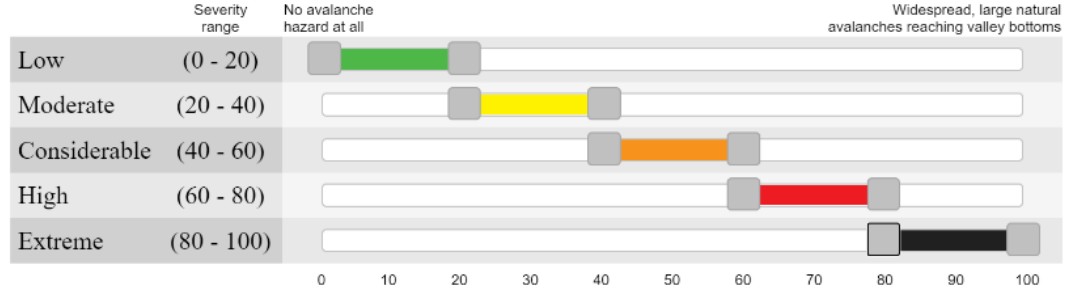

**Figure 2: Screen shot of survey question on avalanche danger perception. The top panel shows the question prompt and starting positions for sliders. The bottom panel shows a completed response with example answers.**




To explore how participants use the danger rating when planning a trip into the backcountry, the survey included a question that asked participants how the danger rating levels typically affect their decision whether to recreate in the backcountry (Figure 3).

a) Starting position

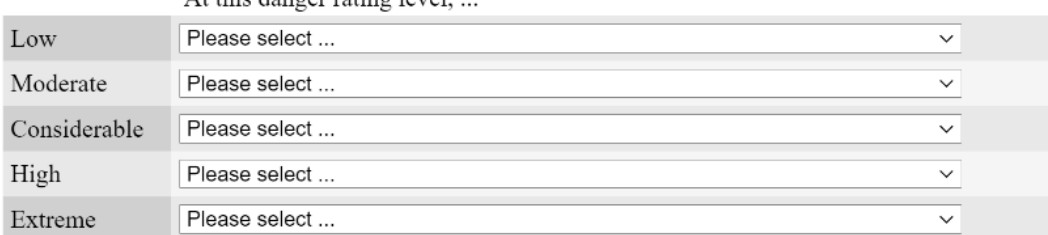


b) Example answer

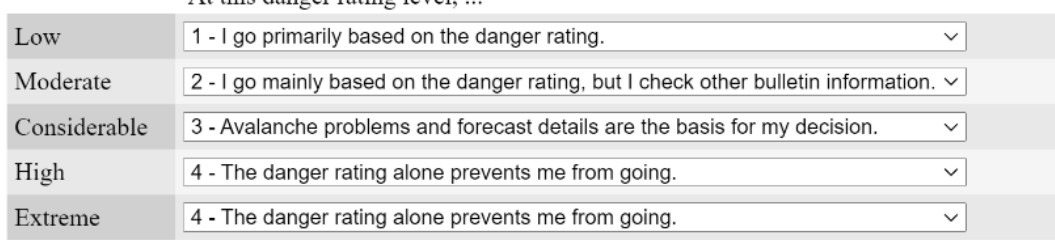

**Figure 3: Screen shot of survey question on typical use of avalanche danger rating levels in trip planning. The top panel shows the prompt and starting position for drop-down options. The bottom panel shows a completed response with example answers.**


For answering this question, participants were given different statements describing possible danger rating use cases similar to the format of the avalanche bulletin user type question. The four statements participants could choose from for each level of the danger scale were:

At this danger rating level, …





1.  I go [into the backcountry] primarily based on the danger rating.

        2.  I go [into the backcountry] mainly based on the danger rating, but I [also] check other bulletin information.

        3.  Avalanche problems and forecast details are the basis for the decision [to go into the backcountry].

        4.  The danger rating alone prevents me from going [into the backcountry].

The first three statements describe a progression where the decision to go into the backcountry relies increasingly on more

advanced avalanche safety information and the danger rating itself loses importance in the decision-making processes. The

fourth statement represents a situation where the danger rating itself is viewed as the deciding factor for not going into the

backcountry at all.

The survey also included a series of multiple-choice questions to collect background and demographic information on

participants including age, self-identified gender, country of residence, primary winter backcountry activity, years of winter

backcountry experience, average number of days spend in the backcountry each winter, and level of completed avalanche

awareness training (Table 2).

**Table 2: Background variables with response options. Abbreviations/labels are provided in brackets.**

| Background variable | List of response option |
|---|---|
| Primary backcountry activity | Backcountry skiing/snowboarding (BC), snowshoeing (SS), snowmobiling (SM), out-of-bounds skiing from resorts (OB), ice climbing (IC), backcountry skiing/snowboarding accessed with a snowmobile (SMBC) |
| Avalanche awareness training | None, seminar/classroom (Seminar), Introductory recreational (Intro), Advanced recreational (Advanced), professional training (Prof) |
| Years of backcountry experience | 1st year, 2-5 years, 6-10 years, 11-20 years, 20+ years |
| Average number of days of backcountry recreation per winter | 1-2 days, 3-10 days, 11-20 days, 21-50 days, 50+ days |
| Avalanche bulletin user type | Type A, B, C, D, E, F |
| Age | Under 20, 20 to 24, 25 to 34, 35 to 44, 45 to 54, 55 or over |
| Self-identified gender | Male, female, non-binary/third gender, prefer to self-describe, prefer not to say |
| Country of residence | United States or Canada |

**2.2 Survey deployment and analysis dataset**

The survey was available for participation from March 23 to May 31, 2019, and a link to the survey website was distributed

extensively on social media and was also displayed on the websites of several avalanche forecasting centers across North



America. To further incentivize participation, those who completed the survey before May 15, 2019, were entered into a cash prize draw.

During the two-month period when the survey was available, 4690 individuals started the survey. Prior to analysis, 1355 incomplete surveys were removed from the analysis dataset, which represents a 28.9% drop out rate. In addition, we removed responses of participants whose residence was outside of North America, who took less than 10 minutes to complete the survey, whose primary activity does not involve exposure to avalanche hazard (e.g., trail running), and whose avalanche training level we were unable to confidently classify as none, introductory, advanced, or professional level.

The final number of survey participants included in the analysis dataset was 3195. While most of the analysis sample (75.6%) reported to primarily participate in backcountry skiing or snowboarding, snowshoeing (7.6% of sample), mountain snowmobiling (6.0%), out-of-bounds skiing (5.1%), ice climbing (3.4%), and sled-accessed backcountry skiing (1.8%) were also chosen as primary winter backcountry activities. Most respondents (73.0%) identified as male and 25.1% as female. The United States was the residence for 54.6% of the sample, and 45.2% were from Canada. Participants' ages ranged from younger

than 20 years to older than 55 years with the age category 25-34 forming the largest group (38.8% of sample), followed by 35-44 (22.6% of sample). Backcountry experience was indicated both by how many days per year a participant typically spends in the backcountry and by how many years of experience the individual had accumulated. Days per year ranged from 1-2 days per winter to more than 50 days per winter (modal response 21-50 days per winter with 30.1% of the sample). Years of experience varied from those in their first year to those with decades of experience (modal response 2-5 years with 34.4% of

the sample). The level of completed avalanche awareness training also varied considerably in our sample: 17.8% had no formal training, 47.6% had introductory recreational training, 19.3% had advanced recreational training, and 15.1% had training aimed at aspiring avalanche professionals. The most common self-identified bulletin user type was Type E (45.4% of the sample), while 1.3% of the sample identified as Type A.

**2.3 Analysis approach**

We conducted our entire analysis in the R statistical environment (R Core Team, 2022) and started with standard descriptive statistics to describe the nature of the analysis dataset and explore the relationships between different variables. We used Pearson chi-squared tests, Wilcoxon rank-sum tests, or Kruskal-Wallis tests depending on whether the variables of interest were categorical or ordinal, and whether we were comparing two or more groups. Unless stated otherwise, we used a p-value threshold of 0.05 to determine whether differences are statistically significant. However, minute differences that cannot be

interpreted meaningfully were disregarded even if they were statistically significant.

The three main survey questions of interest—the recall, perception, and use of the danger scale—were presented to survey participants who use the bulletin (i.e., Bulletin User Types B-F). This resulted in an original analysis dataset of 3130 responses for these questions. The general analysis approach for these questions included two steps. First, we identified common response patterns using an approach tailored to the question specific response format. In the second step, we related the identified



response patterns to participants' avalanche safety training and backcountry experience to better understand the sources of the observed differences.

### 2.3.1 Response patterns in recall question

To better understand participants' ability to recall the danger scale, we manually examined their free-form text responses in three different ways: a) how many terms a participant recalled, b) which danger rating levels were recalled, and c) how well
participants recalled the entire scale in the correct order. Our assessment of how many terms a participant recalled was only concerned with the number of terms, regardless of whether they were the correct terms. To assess participants' recall of individual levels, responses were graded by whether the participant recalled each term of the danger scale, regardless of whether the terms were in the correct order. Finally, to assess the recall of the entire scale, responses were graded by whether the participant correctly identified five danger ratings by the right terms and placed them in the correct order. Incorrect
responses were categorized to identify common errors.

For our analyses, the standard colours and numbers were considered acceptable substitutes for the danger rating terminology. "Very high" was accepted as "Extreme" if not used in combination with "Extreme". Responses that indicated participants skipped the question or did not understand it (e.g., relaying information on current conditions or providing only the first and last terms of the range) were removed.

### 2.3.2 Response patterns in ordering of signal words question

The responses of survey participants who self-identified as Bulletin User Type A to the ordering of signal words question were graded on their ability to correctly place all five terms in the correct order. Incorrect responses were categorized to better understand the most common errors.

### 2.3.3 Response patterns in perception question

To identify common patterns in the responses to our danger scale perception question, we used a latent class mixed effect model, an analysis approach also known as growth mixture models (Muthén and Muthén, 2000). These types of models combine the capabilities of mixed effects models that account for correlations that emerge from repeated measure designs (Harrison et al., 2018; Zuur et al., 2009) with person-centered latent class or mixture models that can identify the presence of unobserved (i.e., latent) subpopulations and describe them with separate but simultaneously estimated regression models (e.g.,
Collins & Lanza, 2010; Lazarsfeld & Henry, 1968). A complete description of latent class mixed effect models is beyond the scope of this paper, but interested readers are referred to Jung and Wickrama (2008), and van der Nest, Lima Passos, Candel, and van Breukelen (2020) for more details.

To apply this analysis approach to our dataset, we viewed each participant's minimum and maximum severity estimates for each danger rating level as a separate observation and regressed the severity ratings against the danger rating levels. The
resulting regression line roughly goes through the center points of the severity ranges of each danger rating level and therefore



describes how participants perceive the shape of the danger scale. To allow for a variety of shapes to emerge from the analysis, we included both a linear and a quadratic predictor for the danger rating level in the model. A positive parameter estimate for the quadratic term shows that the increase in severity between levels is perceived to increase as one goes up on the scale (i.e., the curve steepens), whereas a negative parameter estimate indicates that the curve flattens out. A quadratic term that is not

significantly different from zero indicates a straight linear relationship between the danger rating level and the perceived severity. To also shed light on the size of the severity ranges, we included an additional predictor for each danger rating level in the regression model. With the minimum and maximum values for a specific danger rating level coded as -0.5 and +0.5 respectively (0 for all other danger rating levels), the resulting parameter estimates provide a direct estimate of the range size. Hence, the fixed effects included in our analysis describe the severity of the avalanche conditions as:

$$\beta_1 DR + \beta_2 DR^2 + \beta_3 Rng_L + \beta_4 Rng_M + \beta_5 Rng_C + \beta_6 Rng_H + \beta_7 Rng_E \qquad (1)$$

with DR being a numeric representation of the danger rating level (Low: 0; Moderate: 1; Considerable: 2; etc.), $Rng_x$ representing whether an observation represented the minimum (-0.5) or maximum (0.5) value of a danger rating level range (0 for all other danger ratings), and $\beta_x$ the regression parameters. Coding Low as zero and omitting a separate intercept ensures that all regression lines start at the center point of the severity range of Low. To account for the repeated measure design,

participant ID was included as a random effect, and we included all the main effects in the mixture model estimate, which means that both the shape of the regression line and the width of the ranges were considered for dividing the sample into different groups. The output of the analysis consisted of parameter estimates for a finite set of danger scale perception patterns, and membership probabilities for each study participant. Readers interested in the full details of the model specification are referred to the provided R code of the analysis (Haegeli et al., 2022).

We used the `hlme()` function of the `lcmm` package (Proust-Lima et al., 2017, 2021) for our analysis, and we ran the procedure nine times to first estimate a model with only a single class and then models with two to nine latent classes. Initial iterations of our model estimations highlighted groups of participants whose response pattern clearly showed that they did not use the sliders as intended. This included participants who only moved one of the sliders (minimum or maximum) for all danger rating levels. To minimize the impact of user error on our results, we removed individuals with these types of responses from the

analysis. Furthermore, we only included participants in the analysis whose severity midpoints grew monotonically to avoid any spurious responses. Once the dataset was clean, we computed our final model estimations. Our evaluation of the models followed the guidance of Nylund-Gibson and Choi (2018) and included model fit statistic, such as the Akaike Information Criterion (AIC; Akaike, 1974) and the Bayesian Information Criterion (BIC; Schwarz, 1978) with smaller values indicating better model fit. However, we also considered classification diagnostics (e.g., average assignment probabilities), as well as the

interpretability and utility of the estimated models for the research question. Given our large sample size, even minor differences in parameter estimates can emerge as statistically significant even though they are practically not meaningful. Hence, the selection of the final model included considerable judgment from the research team. We assigned each participant to a danger scale perception pattern using the largest membership probability.



### 2.3.4 Response patterns in use questions

Similar to the analysis of the perception question, we used a latent class approach for identifying common patterns in participants' responses to the danger rating use question. However, since the five observed response variables are ordinal, the poLCA() function of the poLCA package (Linzer and Lewis, 2011) was more appropriate for this analysis. In comparison to the latent class mixed effect models described in the previous section, a polytomous variable latent class analysis stratifies the sample into a finite number of patterns directly based on the observed ordinal response variables without estimating regressions

in the process. The output of this analysis consists of sets of probabilities that describe the chance of observing each response for each variable in an identified response pattern (i.e., class-conditional marginal frequencies) as well as membership probabilities for each study participant.

Like in the perception analysis, we only included participants in the analysis whose answers to the use question grew monotonically with the danger rating level. This removed response patterns where participants treated higher avalanche danger

rating levels more liberally than lower levels, which we considered unreasonable. Once the dataset was clean, we estimate solutions with two to nine latent classes. We evaluated the fit of the estimated models using the AIC, BIC, classification diagnostics, as well as their interpretability and utility for the research question.

### 2.3.5 Relating response patterns to background variables

We used conditional inference trees (CTrees; Hothorn et al., 2006), a classification tree algorithm based on statistical

hypothesis testing, to shed light on how the observed response patterns in the danger scale recall, perception, and use questions relate to the background, training and experience of survey participants. The CTree algorithm employs series of permutation tests to partition a dataset into smaller and smaller subgroups along splits in the predictor variables that produce children nodes whose distribution of the response variable are maximally different from each other (Hothorn, Hornik, & Zeileis, 2006). The splitting process repeats until the algorithm can no longer find any statistically significant relationship according to the

specified p-value threshold (default value: 0.05). Once the splitting process is complete, the terminal nodes at the end of each branch contain a distribution of the dependent variable that exhibits minimal variation within the node and maximum variation to the immediately adjacent neighbouring node.

In our CTree analyses, we used a consistent set of predictor variables to explore the relationships between users' background and their danger scale recall, perception, and use. This set of predictor variables included users' primary backcountry activity,

country of residence, level of avalanche training, years of backcountry experience, and average number of days of backcountry recreation per year. We use 0.05 as the p-value threshold for all CTree analyses.



## 3 Results

### 3.1 Recall of danger scale levels

Of the 3130 survey participants who were presented with the danger rating recall question (bulletin user types B-F who use
the danger scale at least rarely), 170 were eliminated from the analysis because of missing or incoherent answers. Of the
remaining 2960 meaningful answers, most participants (78.1%) provided five terms (not necessarily in the correct order),
16.2% provided four terms, 3.6% provided three terms, and 1.3% provided various other numbers of terms. Eighteen
participants (0.6%) explicitly indicated that they did not remember any levels of the danger scale.

When analyzing which terms were recalled, regardless of order, participants recalled Moderate (listed by 97.0% of
respondents) significantly more often than the other levels. Considerable was the next most often recalled term (92.7% of
respondents) followed by Low (91.8%). Extreme and High were recalled significantly less frequently than the other levels
(83.3% and 86.2%, respectively; Pearson chi-squared test: p < 0.001).

Slightly more than two thirds of participants (68.9%) recalled all five terms of the danger scale using the correct terms and in
the correct order. The most common patterns among people who did not recall the entire scale correctly were omitting High
(4.7%), Extreme (3.8%), or Considerable (1.6%); reducing the scale to Moderate-Considerable-Extreme (0.9%); and
incorrectly identifying the more severe danger ratings (e.g., Severe instead of Extreme (0.9%) and Severe-Extreme instead of
High-Extreme (0.9%)). Of those who provided four terms, the most common mistakes were omitting High (29.0%), omitting
Extreme (23.1%), omitting Considerable (10.0%), and omitting Low (4.4%).

The CTree analysis examining the influence of background variables on participants' ability to recall the danger scale correctly
(i.e., all five levels in the right order) included 2867 responses as not all participants provided relevant background information.
Our analysis revealed that avalanche awareness training and number of backcountry days per season were both significantly
associated with participants' performance (Figure 4). Avalanche education formed the first split in which participants with
advanced recreational or professional level training more likely to recall the scale correctly than those with lower training
(p < 0.001). For all levels of training, participants who spend more days in the backcountry each winter (i.e., who are more
engaged in their activity) performed better. Primary backcountry activity resulted in two final splits. First, among participants
with no training who spend more than 10 days in the backcountry each winter, mountain snowmobilers and snowshoers
performed significantly worse than the other activities (p = 0.028). The second activity split separated ice climbers from other
activities among professional trained participants who spend more than 20 days in the backcountry each winter (p = 0.007).
While ice climbers performed more poorly, it is important to note that there were only ten included in the resulting terminal
node.

Overall, non-ice climbers with professional level training and more than 20 backcountry days a season (Node 19) performed
the best, with 93.0% (357 out of 384) of the participants assigned to this node recalling the danger scale correctly. At the other
end of the spectrum, only 32.4% (59 of 182) of participants with no training and 10 or fewer days per season (Node 4) correctly
recalled the entire danger scale. This performance was closely followed by Node 7, in which 36.5% (31 of 85) participants




recalled the full danger scale correctly. This group represents mountain snowmobile riders and snowshoers without training who spend more than 10 days in the backcountry each winter.

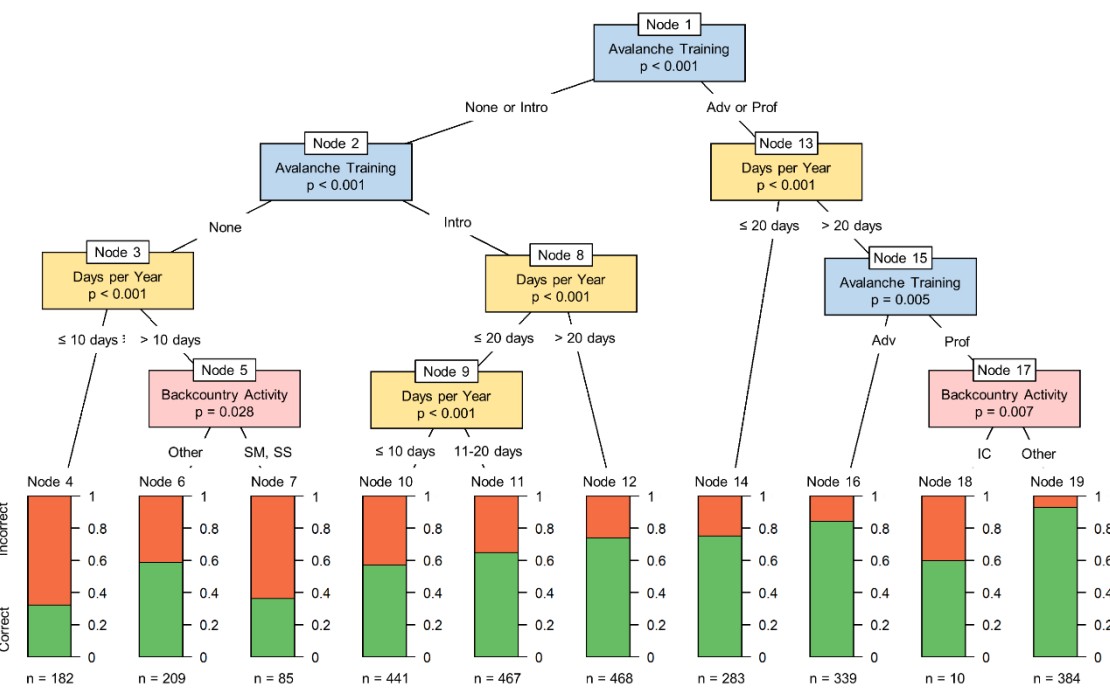

**Figure 4: Results of CTree analysis of danger scale recall response patterns (SM: mountain snowmobiler; SS: Snow shoer; IC: ice**
**climber)**

## 3.2 Order of danger levels

There were 42 Bulletin User Type As directed to the ordering question, and 41 of those participants began answering the question. Of these participants, four people assigned the same term to multiple levels (e.g., assigned Moderate to two levels),
and two people did not complete the question, leaving 35 complete responses for analysis. Of those who completed the question, 26 participants (74.3%) placed the terms in the correct order. In the nine incorrect responses, Considerable was incorrectly placed within the scale seven times: five participants (14.3%) reversed High and Considerable, and two participants (6.0%) reversed Considerable and Moderate. The final two errors were a seemingly random order of terms.



### 3.3 Perception of danger scale

From the 3130 people who began the perception question, 446 responses were removed for incorrectly using the sliders (e.g., not moving one or both sliders), or for failing to indicate a monotonic increase in severity midpoints. The final analysis dataset for this question included the responses from 2684 participants.

Steadily decreasing AIC and BIC values for the latent class mixed effects models with two to nine latent classes indicated that the regression analysis was able to continuously identify new groups with distinct danger rating perceptions. However, while

the parameter estimates for the different classes continued to be significantly different from each other, the practical differences between the regression became irrelevant. Hence, using interpretability as the primary guide, we determined that seven was the optional number of clusters for our avalanche danger perception analysis (Figure 5 and Table A1). Each pattern is characterized by the typical ranges for each danger rating level and a regression line that goes through the centre point of each range.

Almost half of the sample (46.2%) was assigned to **Class 2**, who perceive the danger scale most linearly. The next most common class was **Class 1**, with 24.7% of participants assigned to it and its danger rating curve being the most convex of all the classes. A convex shape indicates that the survey participants assigned to this class perceive the differences between the danger rating levels to decrease as one moves up the scale. The next largest class was **Class 4**, which represented 15.1% of the sample. The participants assigned to this class also perceive the danger rating scale to have a slightly convex shape, but the

most distinguishing feature of this class is the wide severity range for Considerable. The last of the substantial classes is **Class 7**, which represented 7.2% of our sample. Class 7 is the only class whose participants perceive the danger scale to be non-linear with a concave shape. The concave shape indicates that survey participants assigned to this class perceive the differences between the danger rating levels to increase as one moves up the scale. The concave shape is accompanied by increasing danger range widths.

The remaining three classes represent much smaller proportions of the sample. Participants in **Class 5**, which consisted of 3.3% of the sample, perceive the scale to be relatively linear, similar to Class 2, however the ranges of the danger ratings in Class 5 are distinctly wider and have more overlap. **Class 6** represents just 1.9% of the sample. Similar to the participants assigned to Class 1 and 3, this class of respondents perceive the danger rating scale to have a convex shape with diminishing differences between the danger levels as one moves up the scale. Class 6's most distinguishing features are its large intercept

and its very broad ranges at the lower end of the danger scale, which indicates that these survey respondents perceive the danger scale to have elevated severity at these lower levels. The smallest class, **Class 3**, represented just 1.6% of the sample. The shape of the danger curve is convex, similar to Class 1. However, Class 3's most distinguishing features are the exceptional low values for the midpoints of the danger rating levels and its very broad severity ranges, especially at the upper end of the scale.





**Figure 5: Results of latent class mixed effects model analysis of danger rating perception question. Each panel presents the danger rating regression line (black line) and severity ranges for each level (thick vertical lines) from the regression analysis and the severity range distributions of participants' answers associated with this class. Classes are arranged by general shape of the regression lines (starting top left: concave, then linear, then convex). Class sizes are shown above the top left corner of the charts. The numbers included in the panel titles represent the identification number of the latent class.**




The two smallest classes—Class 3 (convex with wide and increasing ranges) and Class 6 (convex with wide and decreasing ranges)—had high class assignment probabilities with median values above 0.950, which means that the observed danger rating patterns are distinct, and participants were assigned with high certainty. The class assignment probabilities were slightly

lower for Class 5 (linear and wide ranges, 0.949), Class 7 (concave, 0.939), and Class 2 (linear, 0.924). The lowest class assignment probabilities were associated with Class 4 (linear and wide considerable, 0.900) and Class 1 (convex and narrow ranges, 0.858). Lower class assignment probabilities indicate that the assignments to these classes were more uncertain, and the response patterns of these participants had some similarity to one of the other classes. Among both Class 4 and Class 1 members, the highest median assignment probability for any other class was for Class 2 (Class 4: 0.059; Class 1: 0.125). This

means that there is some permeability between these classes and the response patterns of several participants sits between these classes.

The dataset for the CTree analysis was 2606 responses since not all participants completed the background questions. The predictor variables that had a significant effect on class assignment were level of avalanche training and number of days of backcountry travel per year (Figure 6). Participants with professional avalanche training had a significantly lower proportion

of members being assigned to Class 2, the basic linear class (36.3% of participants with professional training assigned to Class 2 vs 47.9% of participants with all other levels of training; $p < 0.001$), and a much higher proportion of members being assigned to Class 4, the linear and wide considerable class (20.9% of participants with professional training vs 13.9% for participants with all other levels of training; $p < 0.001$). The number of backcountry days per year had a similar effect to avalanche training. Participants with non-professional avalanche training who spent more than 50 days in the backcountry per

year also had a significantly lower proportion of members assigned to Class 2, the basic linear class (35.5% of participants with more than 50 days assigned to Class 2 vs 49% of participants with 50 or fewer days). However, instead of a higher proportion of Class 4 individuals, this group had a significantly higher proportion of Class 1, the convex and narrow-range class (34.4% of participants with more than 50 days assigned to Class 1 vs 23% of participants with 50 or fewer days).

Those who correctly identified the concave shape of the danger scale (Class 7) did not exhibit a specific user profile; the

participants in this group were a random assortment of engagement, training, experience, and every other explanatory factor we considered.



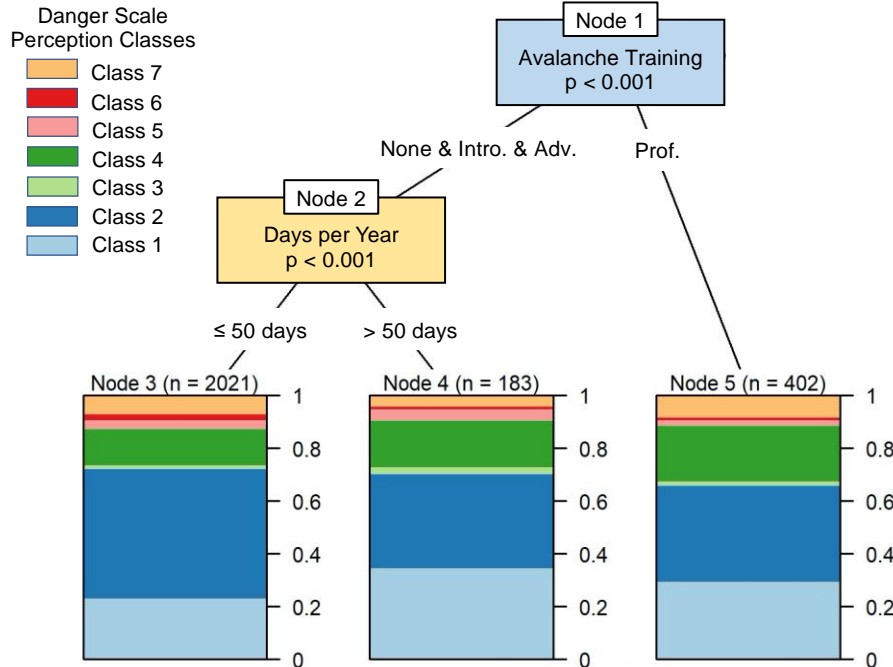

**Figure 6: Results of CTree analysis of danger rating perception question**

## 3.4 Use of danger rating level in trip planning

From the 2705 responses to the danger perception question, a further 126 responses were removed for the use of danger rating analysis because the participants did not answer the question or provided a response that did not increase monotonically. In total, the responses from 2584 participants were available for this part of the analysis.

While the AIC decreased continuously from models with one to eight latent classes, the BIC showed a distinct minimum at six classes. The selection of six as the optimal number of classes was further supported by the fact that the median assignment probabilities were 1 for all the classes and the meaningful interpretation of the emerging cluster solution. Each class is characterized by different proportions of participants' reliance on bulletin information for decision making at each danger level. The stacked bars in Figure 7 represent the conditional (i.e., class-specific) response probabilities at the different danger rating levels. The classes are best described by their most striking patterns:

- **Class 2** participants (5.7% of the sample) primarily rely on the danger rating at both Low and Moderate, avoid the backcountry at Extreme. About half of these participants also avoid the backcountry at High, while the other half generally use other bulletin information to make decisions at High.





- **Class 4** participants (27.9%) rely mainly on the danger rating at Low and Moderate, use other bulletin information at Considerable, and avoid the backcountry at High and Extreme.

- **Class 3** participants (4.4%) rely mainly on danger ratings from Low to Considerable, use other bulletin information at High, and avoid the backcountry at Extreme.

- **Class 6** participants (31.3%) rely mainly on the danger rating at Low, use other bulletin information for Moderate and Considerable, and avoid the backcountry at High and Extreme.

- **Class 1** participants (17.0%) rely mainly on the danger rating at Low and Moderate, use other bulletin information at
Considerable and High, and avoid the backcountry only at Extreme.

- **Class 5** participants (13.7%) rely on other bulletin information for making decisions at all levels; approximately half of this class avoids the backcountry at Extreme.

The order of the classes in the list above and Figure 7 is based on the strong relationship between the latent classes and St. Clair's (2019) bulletin user types (Spearman rank correlation = 0.32; p < 0.001) While more than 50% of the participants
assigned to Class 2 self-identified as Bulletin User Types B and C (18.3% and 32.5% respectively), these proportions are considerably smaller in the patterns that rely less on the danger rating and integrate other avalanche bulletin information. In Class 5, the pattern where avalanche problem information and forecast details is used at every danger scale level, the proportion of self-identified Bulletin User Types E and F are the highest (68.9% and 8.6% respectively).

Overall, 90.7% of participants reported that an Extreme danger rating prevented them from entering the backcountry and 62.0%
of respondents reported staying home at both High and Extreme. In contrast to these no-go decisions, 32.5% of participants stated they would enter the backcountry primarily based on a Low danger rating, and 5.9% of participants would go primarily based on the danger rating at both Low and Moderate. This small percentage for Moderate increased to 43.1% when including respondents who go into the backcountry based mainly on the danger rating, but also check other bulletin information.

A total of 2881 responses were available for the CTree analysis. The most significant predictor variables for the danger rating
use were similar to the variables from the danger perception (Figure 8): days of backcountry travel per year and avalanche training formed seven of the ten identified splits. In addition, backcountry activity, which also emerged as a significant predictor in the analysis of the recall performance, caused two splits. Nationality was an additional significant predictor variable, which was unique to the danger use responses. Surprisingly, danger perception class membership did not have an obvious effect on how people reported using the danger scale. The perception class membership was therefore removed from
the predictor variables to maximize the sample size for the use question.





**Figure 7: Results of latent class analysis of danger rating use question. Each panel presents the response probability for the ordinal response options (i.e., proportion of participants) for the five danger rating levels of one of the identified classes. Class sizes are shown above the top left corner of the charts. The numbers included in the panel titles represent the identification number of the latent class.**

The most significant split in the CTree was between users with more than 20 days of experience per year and those with 20 or fewer days per year (p < 0.001). Those with 20 or fewer days had a greater prevalence of Classes 4 and 6 (do not enter the backcountry at High) and lower prevalence prevalent in Classes 1 and 5 (use other information more often and may go based



on other information at High); the opposite pattern is observed in those with more than 20 days of experience. (In Node 2, Class 4 represents 35.9% and Class 6 is 36.0%; in Node 11, these proportions are 18.9% and 25.2%. For Classes 1 and 5, we see 12.6% and 5.5% in Node 2, and 22.5% and 23.3% in Node 11.)

Avalanche awareness training was of secondary importance for those with more than 20 days of experience ($p < 0.001$): generally, Classes 4 and 6 (who do not enter the backcountry at High) were less prevalent among more highly trained people (Node 19 shows 10.3% and 17.9%, respectively, versus 22.3% and 28.1% in Node 12) and Class 5 (relies on other information) was more prevalent (38.4% versus 17.3% in Node 12); with the exception of Class 6 (which shows no significant differences), the opposite pattern is observed in those with introductory or lower training (Node 12).

Below these two initial splits, primary backcountry activity resulted in two significant splits where mountain snowmobile
riders were separated from the other participants directly into a terminal node. Among participants who spend 20 or fewer days in the backcountry each year, mountain snowmobilers exhibited a significantly higher prevalence of Class 2 (16.1% versus 5.8%), Class 3 (23.0% versus 2.5%) and Class 1 (21.8% versus 12.0%), which are all classes that tend to still go into the backcountry under a High danger rating, but which still rely fairly heavily on the danger rating for their trip planning decisions. Mountain snowmobilers were also separated from other participants in Node 12 on the other side of the tree, which includes
participants who spend more than 20 days in the backcountry per winter and have no or only recreational level training. In this split, mountain snowmobilers had significantly higher prevalence of Class 2 (23.0% versus 4.1%) and Class 3 (13.5% versus 5.1%), but the proportions of Classes 1 and 5 did not differ significantly.

Among the non-mountain snowmobile riders, number of days spent in the backcountry, avalanche awareness training and years of backcountry experience resulted in several additional splits. In addition, nationality was responsible for the final split
of participants who spend 10 of fewer days in the backcountry each year (Node 5). Here, participants from the United States had a significantly higher prevalence of Class 6 members (47.3% versus 30.9%; $p = 0.006$), who generally rely on the danger rating at Low, on other information for Moderate and Considerable, and avoid High and Extreme.



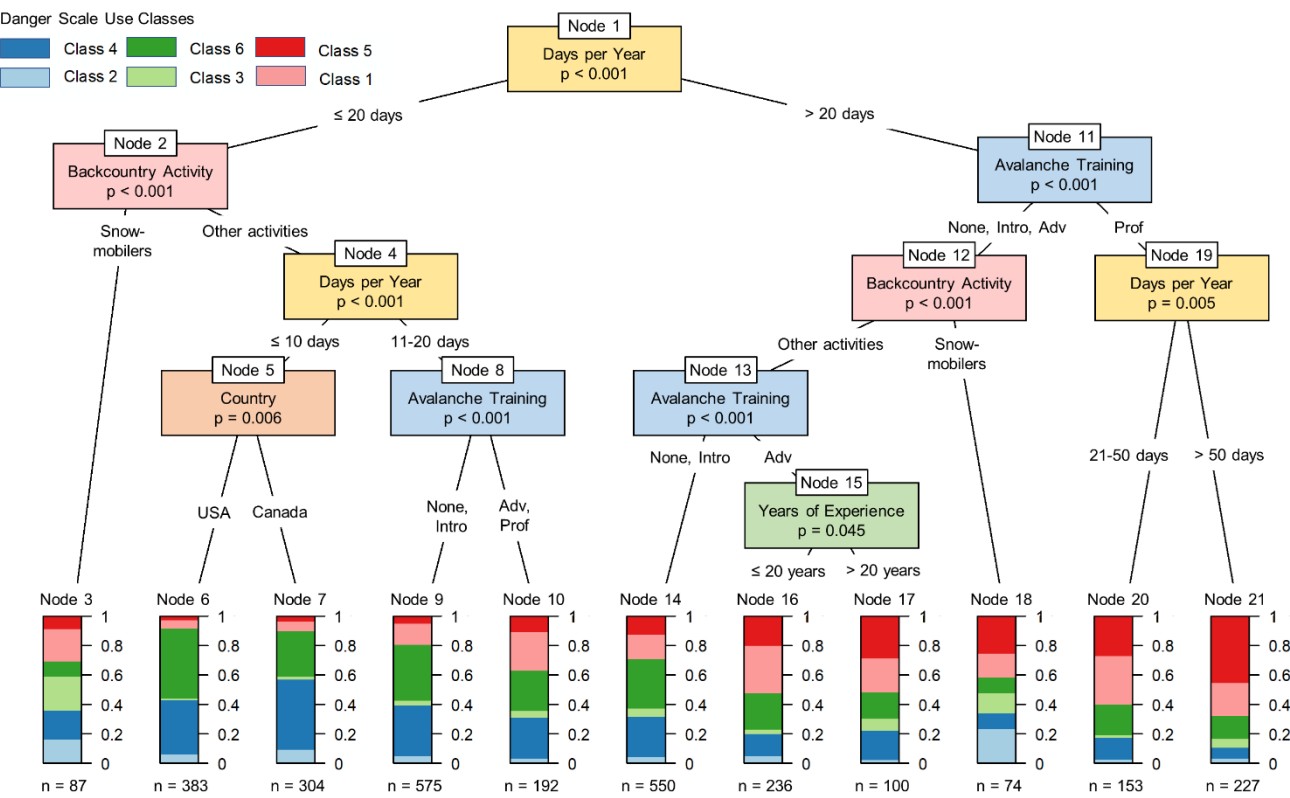

**Figure 8: Results of CTree analysis of danger rating use question**

## 4 Discussion

The results of our study offer a valuable perspective on North American recreationists' understanding and use of the danger scale for general trip planning decisions. At first glance, the key findings include that almost 70% of people correctly recalled the five levels of the danger scale in the correct order; 65% of survey participants perceive the danger scale as a linear scale with or without overlapping ranges; over 90% of participants avoid the backcountry when the danger rating is Extreme, 62% stay home at both High and Extreme; and about a third of participants go out into the backcountry at Low based *primarily* on the danger rating. However, a more in-depth examination of our results provides deeper insight into the effectiveness of the current danger scale.

### 4.1 Understanding of the danger scale

Our results highlight that recreationists' perception of the avalanche danger scale differs substantially from the scientific understanding of the exponential-like increase in hazard between levels most recently described by Schweizer et al. (2020). The predominantly linear interpretation that emerged from our slider question is consistent with the results of other survey





studies that examined recreationists terrain preference as a function of the danger rating using discrete choice experiments (Haegeli et al., 2012, 2020; Haegeli and Strong-Cvetich, 2020). While these studies did not directly ask participants about their perception of the danger scale, the linear patterns in the part-worth utilities provide an indirect measure that aligns with the

results of this study. Aside from the linear perception, a considerable proportion of our sample (28%) associate a convex hazard pattern with the danger scale, and only 7% of the participants expressed a concave pattern, which is closest to the scientific understanding of the scale.

Although the differences between the seven perception patterns are relatively subtle, they all differ substantially from the exponential-like scientific understanding of the scale, which is based on the simultaneous increase of the sensitivity to triggers,

the number of potential trigger locations, and avalanche size from one danger rating level to the next. Studies attempts to quantify the increase in severity have included both hazard-based (Munter, 1997; Schweizer et al., 2020) and risk-based approaches (Pfeifer, 2009; Techel et al., 2015; Winkler et al., 2021) and found a two to four-times increase in severity between danger rating levels. Obviously, such dramatic differences between the scientific understanding of the scale and recreationists' perception have the potential to lead to serious miscommunication about the severity of avalanche hazard.

We suspect several possible reasons for the dominance of the linear perception pattern amongst recreationists. First, many of the most common tools and displays used in North America do not explicitly state the exponential nature of the scale. Examples include the official graphical representation of the danger scale (Figure 1; Statham et al., 2010), the "Introduction to the North American Public Avalanche Danger Scale" video from the National Avalanche Center (National Avalanche Center, 2021), and the Avalanche Canada tutorial focused on the danger rating (Avalanche Canada, 2021). The visual and numerical cues for

interpreting the scale all seem to indicate a linear system, such as each coloured block of the danger scale being the same size and the number of each level increasing by one each step. While there are some resources that explicitly state that the danger scale is non-linear (European Avalanche Warning Services, 2021a; Utah Avalanche Center, 2022), they seem to be less common. Although these different methods of displaying and communicating the scale are not strictly contradictory, they show a predominant linear presentation and inconsistent messaging between educational products.

A second possible reason for the dominance of the linear understanding of the scale could be that it may be the simplest default given the vagueness of the descriptors used in the scale and the challenges that people have understanding these terms. The terms used to indicate the likelihood of avalanches—"unlikely", "possible", "likely", and "very likely"—have been shown to have a broad range of meanings to different people, even when those people have professional level avalanche training (Thumlert et al., 2019). This same difficulty with interpreting terms has been observed across the wider risk communication

community. In psychology, lexical ambiguity in likelihood terms has been identified as a persistent issue for safety and risk assessment (Hancock and Volante, 2020). In medicine, differences in the perceptions of verbal expressions of frequencies between laypeople and physicians have been found to be so great that it rendered the terms unacceptable, leading researchers to recommend that numerical specifications were necessary and preferred (Nakao and Axelrod, 1983). In climate science, public perception of verbal expressions of probabilities improved when paired with a numerical probabilities (Wintle et al.,



2019), and this improvement has been found to span different countries and languages (Budescu et al., 2014). Hence, the persistent issue of ambiguous likelihood terms may be one of the key reasons that makes the danger scale difficult to interpret. Interestingly, those who indicated a concave perception of the danger scale were not a distinct cohort with a specific profile of backcountry experience and demographics, but a rather random mix of participants. Participants who completed professional level training were not more prevalent in the group that expressed the concave perception as anticipated, but rather represented

a higher proportion in the group that indicated a linear perception of the scale with a wider range for Considerable. The same pattern was found amongst recreationists with higher levels of engagement in their activities (i.e., more backcountry days per season). This pattern may reflect the recognition that Considerable can represent a wide range of conditions (e.g., the high-likelihood/low-consequence storm slab avalanche problem situations as well as low-likelihood/high-consequences persistent slab avalanche problem situation), or it may be a sign of respondents' perceived differences in their uncertainty (and therefore

risk) in decision-making between different danger rating levels. While the signs of elevated hazard may become more obvious at High, and therefore make it easier to make conservative decision, the conditions under Considerable may be perceived as vaguer and more error prone. This perception might be supported by the common peak of fatal avalanche accidents at Considerable (e.g., Greene et al., 2006; Harvey & Zweifel, 2008; Jamieson et al., 2010). However, some studies that include the base rates of avalanche danger ratings in their calculations show similar accident rates between Considerable and High

(e.g., Greene et al., 2006; Techel et al., 2015). This inaccurate perception of the danger scale by individuals with professional training has the potential to perpetuate the misconception of a linear scale as avalanche educators share their perspective with their students. In addition, avalanche forecasters' potential inaccurate perception of the danger scale may introduce additional possibilities for miscommunication. However, our study did not specifically examine the danger rating perception of any type of avalanche professional (e.g., forecaster or instructor) to explicitly understand how publishers of the bulletins and avalanche

training providers perceive the scale.

Participants with recreational or no avalanche training who had higher levels of engagement in their backcountry activity were more likely to exhibit a convex perception of the scale. While we do not see an obvious explanation for this pattern, it may be related to participants having more personal experience with the lower levels of the danger scale as most danger ratings assigned by forecasters are between Low and Considerable. Avalanche Canada, for example, forecasted these three danger

ratings over 90% of the time for all elevation bands between 2010 and 2019 (Low 24.2%, Moderate 38.1%, Considerable 28.7%, High 6.0%, Extreme 0.1%, derived from Avalanche Canada public avalanche bulletins archives; (Avalanche Canada, 2021)). Greene et al. (2006) present similar danger rating distributions for the United States, France, and Switzerland. We hypothesize that this higher level of familiarity may lead recreationists to being more confident at distinguishing differences among the lower levels of the danger scale. The more frequent use of the lower levels of the danger scale may also be the

reason that participants showed better recall for these levels. However, the fact that Moderate and Considerable were presented to all survey participants in a slope choice exercise (Finn, 2020) prior to the questions examined in this paper may have also contributed to the good recall performance for these levels.



Regardless of the precise reasons for participants' convex perceptions, an important take-home of our results is that neither higher levels of engagement nor more years of experience result in a more accurate perception of the danger scale. We believe

that this observation is a result of the wicked learning environment (Hogarth, 2001; Hogarth et al., 2015) of winter backcountry recreation, where direct experiences with avalanches are rare and recreationists are given insufficient feedback for developing an accurate perception of the hazard.

## 4.2 Use of the danger scale

Survey respondents' reported use of the danger scale not only provides insight into the role of danger ratings in trip planning,

but also sheds light on the relationship between risk perception and appropriate action. While there are statistically significant differences among the six danger rating application patterns identified by the latent class analysis, four main overarching patterns emerged: a more aggressive pattern with respondents who potentially still go out at High (Classes 1 & 3), a more conservative pattern with respondents who avoid the backcountry at High (Classes 4 & 6), a pattern that relies primarily on the danger rating at both Low and Moderate (Class 2), and a pattern where respondents primarily rely on other bulletin

information for making their decisions about whether to go into the backcountry regardless of the danger level (Class 5). Overall, more than 90% of participants reported staying home at Extreme based on the danger rating alone, and over 60% of participants reported doing the same at Extreme and High.

The primary background variables that determined class membership were participants' engagement in their activity and their level of avalanche training, which were the same background variables that influenced responses in both the recall and

perception questions. In the use analysis, the general patterns showed that less engaged participants and participants with lower levels of training were more prevalent in classes that avoided the backcountry at High, and participants with higher engagement and more advanced training were more likely to use other information to make decisions, particularly at High. Overall, these pre-trip use patterns of the danger scale and their relation to training and engagement seem appropriate, and they are consistent with the reasoning behind the pyramidal structure of the avalanche bulletin, where the most simplified information (i.e., the

danger rating) is presented first in an effort to communicate a general idea of conditions accessible to recreationists with little to no training or experience (Statham et al., 2010). The strong relationship between the observed patterns and the hierarchical typology of avalanche bulletin use introduced by St. Clair (2019) and further described by St. Clair, Finn and Haegeli (2021) also offers an indirect validation of the bulletin user typology.

Although there is a wide body of literature showing that risk perception affects how people respond to risk (Kahneman, 2011;

Loewenstein et al., 2001; Slovic, 1987; Weber et al., 2002), in our study, participants' perception of the danger scale did not emerge as a significant predictor for how people apply the scale in their trip planning. One possible reason why this expected relationship did not emerge could be that the differences between the seven perception classes were relatively subtle, and that our relatively simple question targeting general hazard perception did not consider the nuances of how people form risk perceptions and determine appropriate context-specific behaviours. As described by Loewenstein et al. (2001), Slovic et al.

(2004), Kahnemann (2011), and others, people comprehend risk using two general processes: 1) deliberative or cognitive





processes, which are analytical and calculated; and 2) affective or experiential processes, which are based on emotions, heuristics, and the experiences that create them. Research into natural hazards, such as earthquakes and related tsunamis, has emphasized the importance of direct experiences with a natural hazard to form risk perceptions (Bronfman et al., 2020). Demuth et al. (2016) showed that risk perceptions in hurricane-prone areas can be altered even if someone does not experience

any major tangible effects from an event. Other research in this area has identified that vicarious experiences with unrelated natural hazards can affect how people make sense of a risk message or warning, particularly for an unfamiliar hazard (Sutton and Woods, 2016). We expect that in an avalanche context, where direct experience is relatively rare, recreationists' risk perceptions are being influenced more by analogous or vicarious experiences than direct involvement. In addition to prior experiences, whether personal or vicarious, risk perception is further shaped by how much someone trusts the agency or

authority providing the risk messaging (Griffin et al., 2004, 2013). The situation is further complicated by the risk perception paradox, which highlights that higher risk perceptions in natural hazards do not necessarily spur increased mitigative action (Wachinger et al., 2013). This paradox spans several other contexts, including research about cancer-related worries and risk perception versus vegetable consumption and exercise (Ferrer et al., 2013), and risk perceptions in smoking versus intentions to stop (Klein et al., 2009). In the winter backcountry context, the nature of the wicked learning environment, the voluntary

exposure to the risk, and the exhilarating experience and strong positive emotional rewards associated with the activity may further complicate this relationship (McCammon, 2004).

Given the multitude of factors influencing risk perception and mitigative actions, it is not surprising that choosing appropriate actions based on a danger rating is a challenging task. Although sparse, recreation-related research has demonstrated that people have difficulty identifying appropriate actions using danger scales. Langer et al. (2011) found that the fire danger signs

did not include clear information on appropriate behaviour, and Menard et al. (2018) critiqued rip current warning signs' ability to effectively communicate intended warning messages. The North American Public Avalanche Danger Scale includes guidance for mitigative actions for each danger rating. Although respondents in our survey did not demonstrate a link between their perception of avalanche hazard in the danger scale, they did exhibit reasonable applications of the danger scale in trip planning in relation to their levels of experience and engagement. This finding may indicate that the danger scale effectively

communicates recommendations about appropriate behaviour under different conditions.

### 4.3 Practical implications for avalanche risk communication

Our research has highlighted two main challenges associated with the current scale: inaccuracies in the perception of the scale and difficulties with Considerable. Our results show that the intuitive perception of the scale is linear, while the scientific understanding of the scale shows an exponential-like increase in severity. This discrepancy between perceptions creates the

potential for miscommunication. However, our results also show participants' use of the scale for making the decision whether to go into the backcountry does not seem to be driven by perception. This could potentially mean that the ratings' actionable guidance (i.e., short descriptors provided with each danger rating) are a stronger driver for danger rating use than the perception of the scale. The headline, or short descriptor of conditions typically displayed above the danger rating, may have a similar





effect. But regardless of these potential effects, there is an opportunity to improve alignment between the scientific and public
understanding of the scale.

In addition to the difficulties of varied perceptions, two aspects of our research show that Considerable remains a challenging
condition and a challenging term. Respondents with higher engagement and higher levels of training were more prevalent in
classes with a wider range for Considerable, possibly indicating the wide range of conditions that can lead to a Considerable
danger rating or the difficulty of managing the conditions experienced at this rating. Furthermore, the biggest source of error
in the danger terms ordering question by Type As was failing to place Considerable correctly, indicating that it is a less intuitive
term than the other danger signal words.

The danger scale challenges highlighted by our study are not completely new, and several authors have suggested that a four-
level scale may be more appropriate for recreational risk communication. Some of the justifications behind these
recommendations may assist with aligning the scientific and public understanding of the scale. Using accident data from North
America and Europe, McClung (2000) reasoned that a simpler four-step scale may be more comprehensible for recreationists,
arguing that the current scale should take human perception into account. Conger (2004) supported McClung's proposed four-
level scale because it would shift Considerable out of the middle of the scale and into the more severe end of the spectrum.
Using permit requests and parking data as recreation use data and avalanche bulletins in Glacier National Park (British
Columbia), Eyland (2018) showed that 70-90% of the local recreational use occurred at Moderate and Considerable. He also
found that recreationists tended to treat Moderate and Considerable similarly when deciding to enter the backcountry, and that
recreationists seemed to treat High as significantly more dangerous than Considerable. Eyland therefore recommended
removing Extreme from the scale, which would shift Considerable into the more dangerous half of the spectrum.

Other research has shown that Extreme may not be a practicable rating due to its seldom use and difficulties identifying the
condition correctly. When comparing the predicted danger ratings published in avalanche bulletins with the nowcast
assessments of the following day in three regions in the Canadian Rockies, Statham et al. (2018b) found that Extreme was
forecast just 0.03% of the time in over 3,752 bulletins, and that these few instances were incorrect 89% of the time when
compared to the nowcasts. Low was correctly identified most often (84%), and accuracy fell with each subsequent increase in
danger rating: 71% for Moderate, 69% for Considerable, 55% for High, and 11% for Extreme. Techel and Schweizer (2017)
found similar results in their analysis of danger rating estimates in Switzerland. Also comparing forecasts to nowcasts, they
showed highest accuracy at Low (86%) and lowest accuracy at High (28%). Extreme was not included in this study due to the
small number of instances. The rarity of Extreme and the difficulties associated with correctly identifying it led Statham et al
(2018) to stress the importance of focusing risk messages on information that can be used by the receiver in a meaningful way.
This observation is supported by research from Aitsi-Selmi et al. (2016) that focuses on improving government
communications of complex science in disaster risk reduction strategies using strategies that are "useful, usable, and used". A
danger rating that is hardly used has limited utility and usability in a public risk communication context.

When making recommendations for improving a risk communication tool, it is critical to have a clear understanding of the
target audience (Lundgren and McMakin, 2018). Since the community of avalanche bulletin users is highly diverse, it is



important to properly reflect on which segment of the community depends on the danger rating the most so that the communication tools can be optimized for that audience. The simplicity of the danger rating targets lower-level users, who may not have the training, experience, or desire to pursue, comprehend, and integrate more complex information into their risk management processes. This was explicitly described by St. Clair's (2019) typology of bulletin users, and the responses to our danger rating use question further confirmed that less engaged respondents with lower levels of training are more prevalent in classes that rely more heavily on the danger rating at the lower end of the scale. These users also depend on the danger rating as a threshold for entering or avoiding the backcountry. In addition to these trip planning contexts, the danger rating is often incorporated into decision making frameworks and aids, such as the Reduction Method (Munter, 1997) or the trip planning tool of the Avaluator (Haegeli, 2010), which systematically aid basic judgement and decision making in avalanche terrain and are often recommended as a basic tool for beginner backcountry recreationists. Because the danger rating is so pivotal to these lower-end users, it is critical that the scale be intuitively understood by them.

Our results, combined with these observations about Extreme and the target audience of the danger rating, contribute to the discussion of whether a four-step scale would better serve recreationists. Given that a linear perception of the scale is seemingly the most intuitive, removing a level in the upper end of the scale would move Considerable above the centre point of the scale and thereby increase the perceived severity of the condition. Extreme is rarely used and is often incorrectly forecast, and many lower-end users, who are the primary target audience of the danger rating, do not distinguish between High and Extreme. These users essentially already only use four levels out of the current five and removing Extreme would align the danger scale better with the needs and abilities of those who depend on it the most. More advanced users who may use Extreme as their personal threshold for not entering the backcountry could be warned about the extreme hazard conditions in other bulletin parts such as the headline, the avalanche problem information, or the condition summaries because they have the necessary skills to interpret these products in a meaningful way. Hence, eliminating Extreme from the public avalanche danger scale would likely have little impact on these users.

Despite these potential advantages, there are several substantial challenges with changing the well-established danger scale from five to four levels. Retraining users can be difficult, as shown in previous research illustrating recreationists' tendency to revert to previous practices even after being presented with new definitions for the existing danger rating levels (Ipsos Reid, 2009). Changing the scale would also require avalanche risk communicators to redevelop or adjust many existing products, such as decision aids, online tutorials, educational materials, professional resources, roadside signs, etc. Consistency between regions was one of the driving factors for selecting five levels for the danger scale in North America (Statham et al., 2010), and adopting a four-level scale in North America would create inconsistencies with other regions (e.g., Europe). In fact, confusion resulting from differences in danger rating scales used in European countries in the late 1980s and early 1990s was the impetus for the design of the unified five-level danger scale (Mitterer and Mitterer, 2018). If a four-level recreational danger rating scale were to be pursued, the development would have to be supported with a comprehensive overview of the best practices surrounding each of its components including the signal words, icons, colours, symbols, and the graphical representation of the overall scale to determine how to best portray the new scale to ensure efficient and accurate information




transmission with the target audience. A new four-level danger scale would have to be extensively user tested with the intended target audience before it is introduced to the community to ensure it is truly an improvement.

In lieu of reducing the danger scale to four levels, avalanche warning services may wish to capitalize on existing strengths to improve risk communication strategies for a wider audience. Our research highlighted that survey participants seem to use the current system in an appropriate fashion for deciding whether to go into the backcountry. As explained earlier, we attribute this observation at least partially to the travel advice column in the danger rating definition table (Figure 1). To build on this result, avalanche risk communicators may wish to put even stronger focus on recommended protective actions as they provide user with tangible guidance on what to do under specific conditions. Providing this type of guidance has been recommended

by Mileti and Sorenson (1990) in a general emergency public warning context and by Sutton and Woods (2016) with respect to tsunami warnings. This is consistent with previous recommendations by Klassen (2012) who highlighted the need for developing terrain-based tools and decision aids, particularly to support recreationists in field-based decision making. Linking actions and field-based tools to varying levels of user skill is critical, as previous research has shown that many backcountry recreationists overestimate their ability to apply bulletin information to terrain selection scenarios (Finn, 2020). The

development of these targeted terrain-based tools and decision aids could offer opportunities for creating new products that overcome the shortcomings of the existing danger scale without the need to change the existing scale and the risk of confusing existing users. In addition, improvements in the presentation of the danger scale may further improve users' understanding of the scale. For example, the exponential-like increase in severity could be emphasized in educational materials, the graphical representation of the scale, and the numbers that indicate the danger rating levels. These simple strategies are likely insufficient

for creating drastic improvements, but they provide options for better aligning the public and scientific understanding of the scale over the long term.

## 4.4 Limitations

While this research provides useful insights about how people perceive and use the danger scale, there are limitations to the study that should be considered when interpreting and generalizing the results. First, stated danger rating use may differ from

actual danger rating use, which is a common limitation of survey research. Second, the participants of our survey study do not necessarily provide a representative sample of the full range of backcountry recreationists. Our recruitment strategy aimed to recruit a diverse sample, but our sample is still biased towards backcountry skiers and recreationists with and existing and heightened interest in avalanche safety. This relative dominance of more engaged and advanced backcountry users may have prevented some of the subtleties between lower-level bulletin users from emerging in our results. Future research on avalanche

bulletin users should develop better strategies for collecting a more comprehensive and representative sample of the winter backcountry community.

Several aspects of the survey design may have influenced participants' responses. First, the avalanche bulletin scenario included in the slope choice exercise (Finn, 2020) presented in the survey prior to the questions examined in this paper may have primed participants to remember Moderate and Considerable in the recall question. However, both Low and Considerable





were recalled correctly by almost 90% of participants despite Low not being previously mentioned in the survey. Second, the complexity of the sliders in the perception question might have limited participants' ability to accurately represent their perspective. Third, the survey's focus on trip planning and relatively simplistic danger rating use question do not provide comprehensive insight into how recreationists use danger ratings in their risk management practices, both in the trip planning stage and out in the field. While several studies have examined the application of danger ratings to slope-scale decisions using

discrete choice experiments (e.g., Haegeli et al., 2012, 2020; Haegeli & Strong-Cvetich, 2020), more qualitative research might provide a richer perspective on role of the danger scale in field decision. Having an in-depth understanding of the application of the danger rating to terrain is of critical importance before making any changes to the scale.

**5 Conclusion**

Avalanche bulletins are a crucial source of information for recreationists to plan and carry out informed backcountry travel.

To effectively communicate hazard information to a wide variety of recreationists, avalanche risk communicators must understand how different users amongst their audience are perceiving and using the different components of avalanche bulletins, including the danger ratings. A better understanding of how people with different levels of training and engagement interact with danger ratings can highlight valuable opportunities for making them resonate better and increasing their effectiveness.

We analyzed responses from an online survey to evaluate recreationists' perception of the North American Public Avalanche Danger Scale and how they use danger ratings to determine whether to enter the backcountry. Our results show that most recreationists (70%) correctly recall the five levels of the danger scale in the correct order. The most common mistakes in recalling the danger scale or ordering the signal words were associated with Considerable, High, and Extreme. Nearly 65% of participants indicated a linear perception of the danger scale. Approximately 28% of participants showed a convex perception,

and just over 7% of participants indicated a concave perception. These three major perception patterns indicate that recreationists' understanding of the danger scale differs substantially from the scientific understanding of the exponential-like increase in severity between danger levels. In terms of using a danger rating in a trip planning context, over 90% of participants reported that an Extreme rating would prevent them from entering the backcountry, and more than 60% of participants avoid the backcountry at both High and Extreme. About a third of participants reported primarily using the danger rating to decide

to enter the backcountry at Low.

These results complement existing research on the danger scale. Previous research has addressed the nature of the scale (Schweizer et al., 2020), avalanche forecasters' application of the scale (Statham et al., 2018b; Techel and Schweizer, 2017), and accident distributions according to different danger ratings (Greene et al., 2006; Pfeifer, 2009; Techel et al., 2015; Winkler et al., 2021). Our research fills an important gap in understanding how recreationists, the primary target audience of bulletin

products, interact with danger ratings. Given that danger ratings are more critical for less experienced recreationists who interact with the bulletin in less sophisticated ways (St. Clair et al., 2021), it is important that the ratings resonate with these

types of users and empowers them to make safe backcountry decisions. While the current five-level danger scale seems to serve higher-trained individuals well, Extreme may be an extraneous level for lower-level users, whose trip planning process depends more heavily on the danger rating.

While this study provides a meaningful starting point for a better understanding of bulletin users' perception and use of the avalanche danger scale, there are numerous opportunities for future research. For example, better insight into how danger ratings are applied to terrain choices both in trip planning and while out in the field is critical for getting a more complete picture of danger rating use practices. Furthermore, examining how bulletin users react to danger rating trends (e.g., does changing from Considerable to Moderate elicit the same perception and use as changing from Low to Moderate?) might

provide valuable insight to help avalanche forecasters better understand the effect of their messaging. Combined with the results of this research, these studies may provide valuable lessons on the effectiveness of danger scales for avalanche warning services.

Given that little research has addressed how public perceive and use danger ratings in any type of hazardous environment, we also suggest that similar research projects should be pursued in other hazard contexts. A more comprehensive examination of

hazard scales across other hazard domains might reveal overarching lessons that will help to improve their overall effectiveness.

**Code and data availability**

The data, code, and output for our analysis and the data and code for the figures and tables included in this paper are available at https://doi.org/10.17605/OSF.IO/RTMYX (Haegeli et al., 2022).

**Author contributions**

PH and HF conceptualized the study and designed the survey. AM and PH analyzed the data with assistance from PM. AM prepared the original draft of the manuscript with help from PH. All authors reviewed the manuscript prior to submission.

**Competing interests**

PH is a member of the editorial board of *Natural Hazards and Earth System Science*. The authors have no other competing

interests to declare.

**Acknowledgements**

The authors thank Dr. Robin Gregory (Senior Research Scientist, Decision Research), Karl Klassen (Warning Service Manager, Avalanche Canada) and Anne St. Claire (Simon Fraser University) for useful discussions and guidance during the



design of this study. Thanks also go to Avalanche Canada, the Colorado Avalanche Information Center, and the Northwest

Avalanche Center and all other avalanche warning services for their efforts to promote our study among their communities. We also thank all our survey participants without whose contribution this research would not be possible.

We are grateful to the Coast Salish peoples including the Tsleil-Waututh (səl̓ilw̓ətaʔɫ), Kwikwetlem (kʷikʷəƛ̓əm), Squamish (Sḵwx̱wú7mesh Úxwumixw) and Musqueam (xʷməθkʷəy̓əm) Nations, on whose traditional and unceded territories Simon Fraser University and our research program resides. This research was conducted across Canada and the United States, which

include the traditional territories of many other Indigenous Peoples.

**Financial support**

AM received funding for this project from MITACS through the Accelerate program (IT18825). The contribution of HF was supported by Avalanche Canada's Canadian Avalanche Information Distribution System (AvID) project, which was funded by Public Safety Canada's Search and Rescue New Initiative Fund (SAR-NIF). Public avalanche safety research at the Simon

Fraser University Avalanche Research Program is further supported by the Avalanche Canada Foundation and Canada's Natural Science and Engineering Research Council (NSERC) through the Industrial Research Chair in Avalanche Risk Management.




## Appendix A

This appendix presents the parameter estimates for latent class mixed effects model for the danger rating perception analysis.

**Table A1: Parameter estimates for latent class mixed effects model for the danger rating perception analysis**

|  | Concave (Class 7) | Linear (Class 2) | Linear & wide Cons. (Class 4) | Linear & wide ranges (Class 5) | Convex & narrow ranges (Class 1) | Convex & wide, decr. Ranges (Class 6) | Convex & wide, incr. ranges (Class 3) |
|---|---|---|---|---|---|---|---|
| Class size | 194 (7.2%) | 1241 (46.2%) | 404 (15.1%) | 89 (3.3%) | 663 (24.7%) | 50 (1.9%) | 43 (1.6%) |
| Fixed effects[1] |  |  |  |  |  |  |  |
| Intercept | 9.764 (0.522) | 10.102 (0.221) | 13.437 (0.404) | 21.113 (0.836) | 13.859 (0.387) | 34.712 (1.235) | 14.530 (1.087) |
| Linear term | 13.700 (0.497) | 19.032 (0.203) | 21.385 (0.343) | 17.185 (0.685) | 27.613 (0.333) | 19.308 (0.923) | 16.276 (0.968) |
| Quadratic term | 0.775 (0.125) | 0.268 (0.048) | -0.627 (0.086) | -0.811 (0.169) | -1.724 (0.088) | -1.101 (0.225) | -1.397 (0.244) |
| $Width_{Low}$ | 16.021 (0.603) | 17.063 (0.243) | 22.784 (0.489) | 35.667 (1.102) | 19.604 (0.390) | 54.514 (1.590) | 22.344 (1.215) |
| $Width_{Moderate}$ | 24.307 (0.651) | 20.732 (0.249) | 36.358 (0.539) | 51.901 (1.052) | 27.376 (0.431) | 50.740 (1.301) | 39.108 (1.285) |
| $Width_{Considerable}$ | 35.640 (0.797) | 24.336 (0.287) | 45.099 (0.517) | 58.496 (0.912) | 27.873 (0.420) | 42.085 (1.292) | 58.855 (1.304) |
| $Width_{High}$ | 41.168 (0.741) | 23.500 (0.259) | 38.482 (0.496) | 56.584 (0.976) | 19.121 (0.399) | 26.958 (1.484) | 73.680 (1.364) |
| $Width_{Extreme}$ | 45.418 (0.691) | 19.600 (0.308) | 22.355 (0.506) | 45.757 (1.212) | 8.224 (0.354) | 10.846 (1.332) | 82.633 (1.423) |
| Class assignment probability |  |  |  |  |  |  |  |
| Median | 0.939 | 0.924 | 0.900 | 0.949 | 0.858 | 0.954 | 0.979 |

[1] Parameter estimates and standard errors in brackets. The p-values of all parameter estimates are smaller than 0.0001.

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
