# Peer review of "A user perspective on the avalanche danger scale – Insights from North America"

_EGUsphere, 2022_

## Referee Comment (RC1)

**Review of "A user perspective on the avalanche danger scale – Insights from North America", by Morgan, A., Haegeli, P., Finn, H. & Mair, P.**

**General comments**

This paper investigates how different levels of avalanche danger, visualized by the avalanche danger scale, are interpreted and applied by backcountry users. As described by the authors, "snow avalanches are a serious natural hazard […] that can threaten settlements, transportation corridors, critical infrastructure" (row 32-34). In addition, avalanches pose a lethal threat to recreational users of mountainous terrain.

The avalanche danger scale is a color coded 5-level scale that summarizes current avalanche conditions. The scale is ordinal and uses natural language to describe the different levels (1 - low, 2 - moderate, 3 - considerable, 4 - high, 5 - extreme). Although the numbers used (i.e., 1 – 5) suggests a linear scale, the underlying scale is meant to be exponential (i.e., $b \cdot a^{danger\ level}$, where $b$ and $a$ are some constants). Based on Munther's calculations on avalanche activity in Switzerland, the hazard increases two-fold between each level. This suggests that $a = 2$. In addition to the forecasted danger level, bulletins usually also include a main message, a description of the most important avalanche problems, distribution of these, and a snowpack history. The danger scale is commonly placed at the top of the information pyramid.

Many papers that analyze behavior and perception of risk in avalanche terrain include the avalanche danger as a factor. However, these papers usually focus on a few specific danger levels and do not estimate differences in perceived risk at different danger levels. A few papers have investigated if avalanche bulletin readers understand the information in the avalanche bulletin and draw correct conclusions from the information. A general finding is that the danger level alone is not sufficient to make correct terrain choices.

In contrast to previous papers, this paper analyzes how backcountry users perceive the increase in avalanche danger from one level to another. The paper further analyzes if different danger levels affect users' terrain choices differently, and if it is possible to identify different groups, with different perceptions of the danger scale and different behavioral responses. Finally, the authors also test if respondents remember the different avalanche danger levels, their names, and if participants can correctly rank the different levels.

The authors elicit perceived avalanche hazard by asking respondents to provide an interval of risk on a scale from 0 (no avalanche hazard at all) to 100 (widespread large natural avalanches reaching the valley bottom). The danger levels were stacked on top of each other so that respondents could easily compare one danger level to another.

The authors use a categorical scale to evaluate how the respondents use the danger scale to make terrain choices: 1) "I go primarily based on the danger rating", 2) "I go mainly on the danger rating, but I check other bulletin information", 3) "Avalanche problems and forecast details are the basis for my decision, and 4) "The danger rating alone prevents me from going".

The empirical analysis is based on a relatively large sample of backcountry users (N = 3195). The authors use a latent class mixed effects model and conditional inference trees to analyze the data.

Mixed effects models are used to analyze hierarchical data where observations are clustered (and possibly correlated) within groups (e.g., pupils in a school). Latent class mixed models allow researchers to identify latent group structures in a dataset.

In this paper, the authors use latent class modelling to identify different classes (or patterns) in danger scale recall, perceived functional form and use of the avalanche danger scale.

Conditional inference trees, also called unbiased recursive partitioning, is a non-parametric class of decision trees that uses statistical theory to select variables. The method is commonly employed in machine learning. The aim of the model is to identify variables (or rules) that partition the data into groups that are maximally different from each other. The rules are based on the statistical association between the explanatory (or predictor) variables and some outcome.

In this paper, the authors use conditional inference trees to understand how the response pattern in danger scale recall, perception and use questions relate to the background of the respondents. The paper is well-structured and the authors provide a clear and in-depth description of the data and empirical analysis.

**Evaluation:**

Both professional and recreational users of avalanche terrain rely on avalanche bulletins for information about avalanche hazard. It is therefore important to understand how users understand and use the information presented in the bulletins. Previous research shows that a fair share of backcountry users relies heavily on the danger level when they make decisions on where to go in the backcountry. It is therefore interesting to understand how backcountry users perceive and use the different danger levels of the scale. **In conclusion, the topic is relevant and should fit well within the scope of NHESS.**

The analysis of how people perceive and use the avalanche danger scale is novel and worthy of investigation. I am more skeptical about the value of testing if people remember the names of different danger levels. The latent class mixed effect model has not been previously applied in this field of research. It constitutes an interesting approach to gain a more detailed understanding of the data. The Conditional Inference tree method is an interesting approach to understand the association between background variables and the identified response patterns in the latent class analysis. The method can further identify new, interesting groups in the data. However, I have several concerns with the paper, described in detail below.

To make my discussion here meaningful, I first need to point out an error in the text, the authors say that class 7 in figure 5 is concave, and that class 1 and 6 are concave. This is wrong. The function in figure 5a is convex according to the commonly accepted mathematical definition. The functions in figure 5 e and f are concave. Below, I refer to convex as a function that has a slope that increases at higher danger levels, and a concave function as a function where the slope diminishes with the danger level.

**A. Elicitation of perceived hazard at different danger levels.**

*Ambiguity of question asked*
In the survey, the respondents are asked to: "For each of the five danger rating levels, use the two grey sliders to indicate the severity range of the associated avalanche conditions on a scale from 0 (no avalanche hazard at all) to 100 (widespread, large natural avalanches reaching valley bottoms)." It is not clear if the respondents are asked about their experiences of different avalanche hazard at different danger levels or if they are asked about what hazard each level *should* represent. This may seem like a detail, but it may be important for the results. The reason is that the forecasted avalanche danger may be wrong, e.g., a forecasted level 3 that is really a level 4. If the question is understood as "the spread in danger during a forecasted level 3", then we should expect to see overlaps between different danger levels (here, level 3 and 4). If the question is instead meant to capture the conceptual meaning of the danger levels, we should not expect such overlaps. The problem is that different respondents may have interpreted the question differently.

*Bias*
The respondents were asked to choose an upper and lower bound for the hazard at each danger level. I am sympathetic to this approach, because it is probably a lot easier to do this, than to provide an estimate of the exact level. However, this approach also causes problems. In the graphical analysis, the midpoint is used to identify the functional form. However, this presumes that the respondent thinks that midpoint represents the most common level of hazard at a given avalanche danger. This is a strong assumption if the respondents rely on their experiences of the forecasted danger level (i.e., when the hazard can be under- or overestimated). This is most evident for danger level 1 and 5 (a forecasted level 5 can be a real level 4 but never a level 6. Level 1 is similarly censored from below). However, it is also possible that there is bias in how levels 2 - 4 are forecasted. The problem is that we do not know what this bias is, and the approach used to elicit the functional form does not allow us to identify it. This problem could have been avoided if the respondents had been asked to identify a spread around a chosen point on the scale.

*Construct validity*
The aim of the question is to identify the perceived functional form of danger scale. Given that the danger scale is exponential with a base 2, this means choosing intervals around the points $b \cdot 2$ (low), $b \cdot 4$ (moderate), $b \cdot 8$ (considerable), $b \cdot 16$ (high), and $b \cdot 36$ (extreme). A participant who knows that the hazard doubles on each step thus first needs to understand that s/he should choose $b \cdot 2^{danger\ level}$ to find a point estimate on the scale, and then place the spread around this point. This is a challenging task. Since the endpoint of the scale represents a level 5 situation, this means that 100 should be included in the spread around level 5. Similarly, the endpoint for level 1 should include zero. This may lead to a different spread around these levels. Since the scale is exponential, the spread will increase with higher danger levels, unless the respondent is expected to leave gaps in the scale (suggesting that some hazard levels are not forecasted). In conclusion, my fear is that the complexity of the task makes it more into a test of math and graph skills than a test of avalanche hazard perception.

It may be that I am making a big fuss about something that is not a real problem. It is possible that people with expert knowledge of the avalanche danger scale would be able to rate the different danger levels in a way that would produce a convex function. To know if this is the

case, the scale needs to be validated. As of now, we do not know what a "correct" answer looks like, or if it is at all possible to answer in a "correct" way. Unfortunately, the problem with the elicitation method severely limits the value of the analyses of the responses.

**Recommendation:** ask a panel of avalanche forecasters to answer the same question as in the survey. Use this as the "ground truth" and compare it to the responses used in the current dataset. In the expert survey, I would also ask the experts to provide a point estimate to get an idea about if the midpoint can be used as a point estimate for the functional form. To check that the ambiguity of the question (spread in actual hazard during e.g., forecasted level 2 versus which hazard level a level two should represent) does not affect responses, I also recommend to re-run the survey question on a small sample of backcountry users where you clarify what it is that you are after.

**B. Use of the avalanche danger scale**

As mentioned by the authors, the "use" question was relatively crude. It only asks to what degree the decision to go into the backcountry depends on the danger level, and not if the danger level affects terrain choices in the backcountry. It may also be mentioned that category 1 encompasses category 4 (both imply that the respondent rely mainly on the danger scale).

**C. Latent class modelling**

The authors use a latent class mixed effects model. This seems like an adequate approach, as it can identify interesting response patterns in the data. However, as noted by the authors, the model did not converge in the sense that the model fit kept improving at a higher number of latent classes. Instead, the authors stopped the process when the new latent classes did not provide meaningful information. I do not doubt that the research group took great care in analyzing the data and determining the number of latent classes. However, I still fear that they read more into the data than what is actually there. It seems reasonable to divide the responses into convex, linear, and concave functions, with narrow and wide spreads. However, although the classes are significantly different from each other, they are all relatively linear from a practical perspective (eyeballing the graphs, the step size in mid-points between different levels appears to be about 15-20 for all functions). Based on the AIC and BIC, all seven classes should be included in the model, but so should the discarded classes. This makes me wonder how much we gain from including so many classes and how much we should read into the differences.

**Recommendations:**
1. Perception of the avalanche danger scale:
   a. It would help if we knew why the kept classes are more meaningful than the discarded classes. I therefore recommend that a set of discarded classes are presented in an online appendix. These classes should consist of the first e.g., 3-6 classes that improves fit but that the authors deemed meaningless. I would also like to see a short discussion of why the discarded classes have less meaning than the included classes.
   b. Since the differences in functional form are relatively small, it would help if the authors discussed the practical implications of the differences. It would also be interesting to know which analytical functional form the different graphs represent (i.e., how far from an exponential scale are they?).

c.  If the practical implications of the differences are important, I think that you should put more emphasis on the fact that there was no clear pattern among people who perceived the danger scale as convex with narrow spreads.

**D.  Conditional inference trees**

Conditional inference trees offer a novel and interesting approach to understand the association between background variables and response patterns. In this paper, the trees are used to understand groupings according to 1) the share of correct responses on danger level rank question, 2) latent class combinations with respect to the functional form of the danger scale, and 3) latent class combinations with respect to use of the danger scale.

The conditional inference tree is very helpful to understand the association between background variables and the ability to correctly rank the different danger levels. The authors make an admirable effort to explain the results. However, the final nodes of the inference trees for perception of avalanche hazard and use are very complex and difficult to understand. The reason is that the nodes represent different combinations of latent classes. In the case of *use* of the danger scale, the latent classes further represent different combinations of user types. The problem is made worse because the different latent classes are ordered in a way that doesn't help the reader. I understand that the statistical software labels the classes and that the authors use these class names to enable replication. However, it makes it cumbersome to read the text. I had to scroll up and down to recall what the different classes represent, and make notes, to have a chance to interpret the figures.

**Recommendations:**
I think that it would be beneficial if the authors renamed the classes so that the follow a logical order and given names that describes the class (e.g., for perception: class 1 – convex, class 2 - linear, class 3 – concave. For user group; class 1 – rely mainly on danger level at low levels, class 2 – rely mainly on danger level at high levels, class 3 – rely mainly on other information in avalanche bulletin).

Minor points:

1. I am not sure why it is interesting to analyze if people recall the names of the different danger levels. If kept in the paper, I would like to see a clearer discussion about what we learn from the results.
2. Section 3.3. I think that it would help to first describe the different classes (give some intuition for them) and then describe how many participants fall into each class. I suggest to organize the classes into Convex, Linear, and Concave.
3. Figure 5. I would like to see a caption and figure order that is consistent and easy to remember. I found the figure very confusing, because the caption mixed "a" to "f" with the class labels, and the class labels did not follow any logical order (to me as a reader).
4. Section 3.4. I would have preferred to read a description of the order of the different classes before the presentation of the list, as this would have helped me understand the order. If possible, order the classes based on how "correct" the strategy is. If that is not possible, give the classes a name that describes the group. This will help when the reader is trying to understand figure 8.

5. Discussion: I would like to see a discussion of why we should expect to see a relationship between a specific functional form of the danger scale and a certain use of the scale. It is not clear to me if one of the use strategies is more correct than another and if a e.g., convex functional form would imply more or less reliance on the scale relative to other information in the bulletin.

6. Discussion: the authors say that the results indirectly validate the Bulletin user typology. I would like to argue that this is incorrect. Both the user typology and the question about use of the avalanche scale asks how a person uses the avalanche bulletin, i.e., they are two ways to ask the same thing. The results therefore suggest that we can measure self-assessed use with two methods. To validate if the bulletin user typology represents actual differences in use, we would have to look at actual use. I would like to see a more nuanced discussion about this.

7. Discussion: The discussion about risk perception seems misplaced. The questions in the survey ask about the objective risk for widespread very large avalanches and not about ability to mitigate the hazard. Although the discussion is fascinating, I don't think that it belongs in this paper.

8. All datafiles are available on osf. This is great! However, I would also like to see that the questionnaire (in full) was publicly available. Information about the length (average completion time) should be provided in the text, together with information about where in the survey the questions were placed. The latter is important since the task may have been perceived as relatively challenging for the respondents.

---

## Referee Comment (RC2)

Review of "A user perspective on the avalanche danger scale – Insights from North America" by Morgan et al.

The authors present a study based on more than 3000 survey responses on the perception and application of the avalanche danger scale in North America. Analyzing the data using a latent-class mixed effect mixed model (LCME) and conditional inference trees, two key findings emerge:

- Survey respondents perceive the increase in the severity of avalanche hazard with increasing danger level differently (linear increase) compared to scientific findings (exponential increase).
- However, despite this perception, there are strong (non-linear?) differences when asked how a specific forecast danger level impacts the users' decision-making during trip planning. The use of the danger levels varies as a function of the users' avalanche skills and back-country touring experience.

These findings, from my perspective, are robust and provide a relevant and novel insight into the users' perception and self-stated use of the avalanche danger scale and the five levels. This user-centered perspective may provide valuable input for avalanche warning services aiming to optimize the communication of avalanche hazard to the public. The survey design is well introduced. The methods used for data analysis are, as far as I can judge, applicable and are clearly described, providing sufficient detail. The manuscript is well written.

I only have two points, which the authors may want to consider.

The manuscript is rather long. Shortening some sections might help the reader to focus on the key findings. Moreover, I personally would have liked to see a summary of the results with the responses stratified by the bulletin user typology classification scheme as shown in Table 1. I would expect that this would make the key findings more tangible for practitioners interested in the relevant outcomes of this study. I will provide more details on these two points below.

I consider the topic of the study as suitable for publication in NHESS.

**General comments:**

**Section 3 Results**

Each of the **Result sections 3.1, 3.2 and 3.4** starts with the number of responses available for the respective analysis, and introduces the number of significant classes obtained from the LCME model (e.g. 304-308, 350-359, 416-420). While this makes it very transparent, which data was excluded, I wonder whether there would be a way to present this information in a more concise way to make these sections more focused. For instance, could this information be moved to either a small table and/or a short section at the end of the Methods-Section? By doing so, the reader would still be able to find this information, if interested, but could focus more easily on the findings.

**Section 3.1 Recall of danger scale levels**

Consider moving some of the results to a table. This would provide an easy-to-read overview of some of the results and may also allow to shorten the text. For instance, moving the findings on l304-312 to a Table like this:

| Question | Answers (proportions) |
| --- | --- |
| Number of levels? | 5 levels (78%)
4 levels (16%)
3 levels (4%)
… |
| Recall of signal words? | Moderate (97%)
Considerable (93%)
Low (92%)
… |

**Section 3.3 Perception of danger scale**

A large part of this section describes in detail what is shown in Figure 5. While this is certainly a very interesting way of analyzing the survey respondents' perception of the danger levels, I feel that this entire section could be shortened. Firstly, the slopes shown in the plots of the three largest classes (classes 2, 4 and 1 combined 85%) look rather similar even though statistically different. I understand that the authors also consider these variations as rather subtle (e.g., l508, l597). Secondly, these three classes combined seem equally frequent in the three nodes in the CTree analysis shown in Figure 6 (80 to 85%?), suggesting that the variables used to explain differences in how participants perceive the danger scale fail to really differentiate between class membership. My interpretation of these findings is that most respondents understand the scale as a linear one (about 60%), with some respondents having a (slight?) tendency towards a concave or a convex interpretation, and that the variables describing the respondents' skills and experience can't really explain the LCME class membership. The take-home message is that respondents perceive the danger scale primarily as a linear scale, and thus different compared to the scientific interpretation. This is a robust and relevant finding. – I propose to emphasize the key findings, maybe at the end of this section, and to consider shortening this section.

**Summary of results using bulletin user typology**

I feel that it could be beneficial to the reader, and particularly to those who are interested in gaining insights on the different bulletin user groups, to summarize the key results in a short section and a graph/table, potentially like the following figure:

| Bulletin user typology | Recall and order of DL | Perception of DL | Use of DL in trip planning | Background variable (median, mode) | N (%) |
|---|---|---|---|---|---|
| A | x% five levels x% levels in correct order | --- | --- | years of experience, number of days, … | 45 (1.3%) |
| B | x% five levels x% levels in correct order | xx% linear x% concave | Barplot similar to subplots in Fig 5 | years of experience, number of days, … | … |
| C | … | … | Barplot similar to subplots in Fig 5 | … | … |
| D | … | … | … | … | … |
| E | … | … | … | … | 45.4% |
| F | … | … | … | … | … |

Personally, I would have liked to see such an analysis/summary, as it would have allowed me to link the key findings more easily to different bulletin user groups. Sections 3.1, 3.3 and 3.4, and the corresponding explanations in the manuscript provide similar information, though always from the perspective of the CLME class assignment. However, I had trouble linking the class assignment to the user typology shown in Table 1, which I find a helpful and intuitive classification. Therefore, I propose to look at the data from this perspective too. Such a section may also support the statement on l673-675. Furthermore, linking bulletin user class and the most relevant background variables could provide a helpful overview describing the survey respondents (described also on l 185-199).

**4.3 Practical implications for avalanche risk communication**

A reduction of five to four levels is discussed thoroughly in this section. Beside the numerous arguments for and against such a change (l679-803), it might be worth taking up the point that avalanche forecasts in Europe, and thus the avalanche danger scale, not only target recreational users but are also an important source of information for decision-makers responsible for the safety of the public in residential areas or on transportation networks. For these decision-makers, but also the public at large, the two highest danger levels are particularly relevant to communicate these rare, but very dangerous situations clearly.

Furthermore, a different approach is used by the Swiss avalanche warning service. As of this winter, in the published forecasts the danger level will be shown together with a qualifier (-, =, +) indicating where within the level the avalanche hazard is expected (e.g., SLF, 2022a, SLF, 2022b; described for instance in Techel et al, 2022). This development would also go against the discussed reduction to four levels and could maybe be mentioned/discussed?

**Further remarks**

- L16-17: This statement is correct though one could add that this linear perception is also in line with the danger scale being an ordinal scale.
- L38, 102 (and maybe other places): I am not sure if calling the Canadian avalanche warning services *local* is the most appropriate term. I would find *regional* forecasting more suitable. For instance, the European Avalanche Warning Services provide a forecast for a *region*.
- A terminology question: do you call the *danger levels* (Figure 1) in North America also *danger rating levels* (as for instance in title of Section 3.4)? - Consider using *danger level* throughout, if appropriate.
- Section 2.1 Survey Design: The survey design is described clearly and with sufficient detail. As I am not an expert in designing surveys, I can't judge, for instance, what effect the provision of numerical sliders may have on the responses.
- L170: spend → spent
- L185 vs. 207: Maybe check whether there is a typo in one of the numbers: 3195 responses (L185) minus the 42 A-responses (l343) would result in 3153 rather than 3130 (l207).
- L285: estimate → estimated
- L322-324: is there a word missing in this sentence? After "training"?
- L325-330: In case you intend to shorten some sections, maybe the text describing the two final splits in Figure 4 could be omitted.
- Sect. 3.3 and Sect. 3.3: the classes obtained with LCME model are referred to by numbers (1-7, 1-6) using the same color scheme in Fig. 6 and 8. While the legend titles in these figures indicate that classes are different, this fact could maybe be supported by using different color schemes, and maybe by using once numbers and once letters for labeling.
- L425-437: Consider including the proportion of self-identified bulletin user types – maybe the proportion of B and C combined vs. the proportion of E and F combined – to the description of the six classes.
- L444-448: This is, of course, just a personal preference, but maybe consider moving the overall results to the beginning of this section, followed by the detailed analysis.
- L499: It might be worth repeating that the avalanche danger scale is primarily an ordinal scale (l47), with categorical descriptions of the danger levels. A large share of the respondents got the order of the levels right.
- L510-513: Maybe of interest: a recent study exploring numerous observations related to the contributing factors of avalanche hazard and the corresponding increase in the severity of the hazard with increasing danger level is Techel et al. (2022). This study also shows changes within the forecast danger levels.
- 521-523: Just an observation: At least some warning services in the European Alps present the danger scale as a scale which shows an exponential increase. Examples include the websites from the Swiss avalanche warning service (SLF, 2022a), as well as the avalanche warning service in Tyrol-South Tyrol-Trentino (avalanche.report, 2022). However, there are also European warning services where no such presentations can be found. The question you raise, would therefore indeed be an interesting one to answer: do avalanche forecasters and educators themselves perceive this non-linear increase in the severity of avalanche hazard with increasing danger level, and if they do, do they consider it important to communicate? Given the findings on the use of the danger scale, with danger levels having a considerable impact on stated

decision-making during trip planning, how important is it that users have a different perception of the danger scale?
- L525: Another reason for this linear perception is maybe also the fact that the scale is primarily an ordinal scale with categorical descriptions.
- L591-593: As suggested before, it would be nice if you could link this statement to a figure or short section emphasizing the relationship between the bulletin user typology and the results.
- L714-717: maybe worth mentioning that such terrain-based tools are already operational, as for instance the website skitourenguru.ch, where back-country ski touring routes in the European Alps are risk-rated according to the forecast avalanche conditions and the terrain
- L764-765: Maybe add "in North America" after "bulletin products", as this statement would not be true in Europe.
- L773-775: maybe of interest as you are mentioning how trends in forecast danger level are perceived, Terum et al. (2022) address this topic in their study

**References**

avalanche.report (2022): https://avalanche.report/content_files/ava_danger_en.png (last access, 26 Nov 2022)

SLF, 2022a: https://www.slf.ch/style/respimg/f_a8_0177673001669383856_csm_Lawinengefahr_Bandbreite_EN_a55cd0022d.jpg (last access: 26 Nov 2022)

SLF, 2022b: https://www.slf.ch/en/news/2022/11/subdivision-of-danger-levels-in-the-avalanche-bulletin.html?fbclid=PAAaa1bhxJ-S2RhbPMaaBkiTXaOWKB7AFcrmDxOsC-wtq6wt_0HD9C_-fF3jc (last access: 29 Nov 2022)

Techel, F., Mayer, S., Pérez-Guillén, C., Schmudlach, G., and Winkler, K. (2022): On the correlation between a sub-level qualifier refining the danger level with observations and models relating to the contributing factors of avalanche danger, Nat. Hazards Earth Syst. Sci., 22, 1911–1930, https://doi.org/10.5194/nhess-22-1911-2022.

Terum, J. A., Mannberg, A., & Hovem, F. K. (2022). Trend effects on perceived avalanche hazard. *Risk Analysis*, 00, 1– 25. https://doi.org/10.1111/risa.14003

Walcher, M. (2022): personal communication

---

## Author Comment (AC2)

**Responses to Reviewer #1**

**GENERAL COMMENT**

We thank Reviewer #1 for their constructive review and helpful comments. We particularly appreciate to detailed discussion on the validity of our survey questions and the resulting implications for the results. We also appreciate the explicit recommendations for addressing these concerns. While we are unable to fully address all concerns, we hope that our responses are satisfactory.

**RESPONSES TO SPECIFIC COMMENTS**

**1.1 Description of function form as convex versus concave**

**Reviewer Comment:**

*To make my discussion here meaningful, I first need to point out an error in the text, the authors say that class 7 in figure 5 is concave, and that class 1 and 6 are concave. This is wrong. The function in figure 5a is convex according to the commonly accepted mathematical definition. The functions in figure 5 e and f are concave. Below, I refer to convex as a function that has a slope that increases at higher danger levels, and a concave function as a function where the slope diminishes with the danger level.*

**Author Response:**

While the reviewer is referring to the mathematical definition of convex and concave functions (see, e.g., https://en.wikipedia.org/wiki/Convex_function and https://en.wikipedia.org/wiki/Concave_function) we used the more common language (geometric) interpretation where convex is curved outward, dome-like and concave is curved like the inner surface of a bowl (see, e.g., https://en.wiktionary.org/wiki/convex and https://en.wiktionary.org/wiki/concave). This is consistent with the general use of these terms in the avalanche safety community, where they are used to describe the shapes of slopes. While we acknowledge that the reviewer's suggestion is academically correct, we would prefer to stay with our current terminology as it is more consistent with the common use of these terms in our community.

**To address this reviewer concern, however, we will add an explanation when we first describe the shape of the different curves that describes the two different perspectives and explains why we use the terms in this particular way.**

**1.2. Ambiguity of danger rating perception question**

**Reviewer Comment:**

*In the survey, the respondents are asked to: "For each of the five danger rating levels, use the two grey sliders to indicate the severity range of the associated avalanche conditions on a scale from 0 (no avalanche hazard at all) to 100 (widespread, large natural avalanches reaching valley bottoms)." It is not clear if the respondents are asked about their experiences of different avalanche hazard at different danger levels or if they are asked about what hazard each level should represent. This may seem like a detail, but it may be important for the results. The reason is that the forecasted avalanche danger may be wrong, e.g., a forecasted level 3 that is really a level 4. If the question is understood as "the spread in danger during a forecasted level 3", then we should expect to see overlaps between different danger*

*levels (here, level 3 and 4). If the question is instead meant to capture the conceptual meaning of the danger levels, we should not expect such overlaps. The problem is that different respondents may have interpreted the question differently.*

**Author Response:**

We appreciate the detailed examination of our survey question, and we can see how its wording can potentially be interpreted in the two suggested different ways:

- What levels of severity have you seen in the field under different danger rating levels? (i.e., personal experience)
- What do you think should the severity level be under different danger rating levels? (i.e., conceptual understanding).

Our question was actually targeted at the personal perception of the scale: "When you think about the avalanche danger rating scale, what levels of severity do you personally associate with the different levels of the scale?". Hence, we were interested in participants' personal conceptualization of the scale, which integrates both their conceptual understanding of the scale and their personal experience in the field into their personal perception. Our hypothesis was that it is this perception of the severity of the conditions that drives their decision process whether to travel in the backcountry or not.

If we had the chance to ask the question again, we might revise the wording of the question to make it clearer and ensure a more consistent interpretation of the question. However, we believe that our rather neutral wording was adequate and the differences between these different interpretations are too subtle to affect our main insights about how people perceive the danger scale. While the differences would be interesting to examine academically, the format of our question and subsequent analysis does not seem precise enough to do this in a meaningful way.

In our analysis, we tried to highlight the key patterns relevant for extracting practical implications and avoid pushing the statistical analysis too far. We believe the subtleties in the interpretation of the survey question likely do not affect the main patterns that emerged in our analysis. **However, to address this reviewer concern, we will discuss the different possible interpretations of the survey question and their potential impact on the results in the limitation section of the manuscript.**

**1.3 Bias**

**Reviewer Comment:**

*The respondents were asked to choose an upper and lower bound for the hazard at each danger level. I am sympathetic to this approach, because it is probably a lot easier to do this, than to provide an estimate of the exact level. However, this approach also causes problems. In the graphical analysis, the midpoint is used to identify the functional form. However, this presumes that the respondent thinks that midpoint represents the most common level of hazard at a given avalanche danger. This is a strong assumption if the respondents rely on their experiences of the forecasted danger level (i.e., when the hazard can be under- or overestimated). This is most evident for danger level 1 and 5 (a forecasted level 5 can be a real level 4 but never a level 6. Level 1 is similarly censored from below). However, it is also possible that there is bias in how levels 2 - 4 are forecasted. The problem is that we do not know what this bias is, and the approach used to elicit the functional form does not allow us to identify it. This*

*problem could have been avoided if the respondents had been asked to identify a spread around a chosen point on the scale.*

**Author Response:**

We agree with the reviewer that our analysis approach is built on the assumption that the midpoint between upper and lower bound is meaningful for the analysis of the functional form of the scale. While we understand the logic of reviewer's concern, we do not believe that it matters for the objective of our study. As discussed above, we are interested in the overall perception of participants' perception of the scale. Using the midpoint is the simplest (and therefore the most free of additional assumptions) way for examining the functional form. Our objective was not to examine the quality of the danger rating assessment or explore the distributions of severities under different danger rating levels.

While we appreciate the suggestion of adding an additional slider for the center point of the distribution, we think that it would have made a complicated question even more complicated and cumbersome to complete. We are uncertain whether this would have resulted in more reliable and insightful results. We acknowledge that there might be better/other formats for asking participants about their perception of the danger scale, but our format was a first attempt of tackling this question and provide some basic insight.

**Similar to the previous concerns, we can address this concern by discussing the limitations of our survey instrument and its implications for the analysis in more detail.**

**1.4 Construct validity**

**Reviewer Comment:**

*The aim of the question is to identify the perceived functional form of danger scale. Given that the danger scale is exponential with a base 2, this means choosing intervals around the points (low), (moderate), (considerable), (high), and (extreme). A participant who knows that the hazard doubles on each step thus first needs to understand that s/he should choose to find a point estimate on the scale, and then place the spread around this point. This is a challenging task. Since the endpoint of the scale represents a level 5 situation, this means that 100 should be included in the spread around level 5. Similarly, the endpoint for level 1 should include zero. This may lead to a different spread around these levels. Since the scale is exponential, the spread will increase with higher danger levels, unless the respondent is expected to leave gaps in the scale (suggesting that some hazard levels are not forecasted). In conclusion, my fear is that the complexity of the task makes it more into a test of math and graph skills than a test of avalanche hazard perception.*

*It may be that I am making a big fuss about something that is not a real problem. It is possible that people with expert knowledge of the avalanche danger scale would be able to rate the different danger levels in a way that would produce a convex function. To know if this is the case, the scale needs to be validated. As of now, we do not know what a "correct" answer looks like, or if it is at all possible to answer in a "correct" way. Unfortunately, the problem with the elicitation method severely limits the value of the analyses of the responses.*

***Recommendation***: *ask a panel of avalanche forecasters to answer the same question as in the survey. Use this as the "ground truth" and compare it to the responses used in the current dataset. In the expert survey, I would also ask the experts to provide a point estimate to get an idea about if the midpoint can*

*be used as a point estimate for the functional form. To check that the ambiguity of the question (spread in actual hazard during e.g., forecasted level 2 versus which hazard level a level two should represent) does not affect responses, I also recommend to re-run the survey question on a small sample of backcountry users where you clarify what it is that you are after.*

**Author Response:**

We appreciate the concern about the complexity of the question, and we agree that there might be other ways to tackle this question.

Given that the danger scale level is a human judgment, and recent studies (e.g., Techel et al., 2018) have shown that there is considerable variability in the application of the scale, we think it is unrealistic to say that "the danger scale is an exponential scale with a base of 2" and accept that as the single truth against which the perception of bulletin users should be tested. We think it is unrealistic to expect that participants can provide this "correct" answer in a survey question and it would not necessarily meaningful. Our question was not "What is the scientific functional form of the avalanche danger scale?". Instead, we wanted to shed an initial perspective on how avalanche bulletin users in North America perceive the avalanche danger scale.

During the design of our survey, we experimented extensively with several different formats for this question. One possible alternate format could have been the following:

*In this question, we are interested in your perception of the general relationship between the danger scale levels and the severity of the avalanche conditions. As we go up on the scale, the difference in the severity of conditions between danger rating levels …*

- *… increases (i.e., the difference between considerable and high is larger than between moderate and considerable).*
- *… remains the same. (i.e., the difference between moderate and considerable and considerable and high is the same).*
- *… decreases (i.e., the difference between considerable and high is smaller than between moderate and considerable).*

Our testing showed that this question format is not necessarily easier to understand and still requires considerable "mathematical thinking". Furthermore, it would have not provided any information about how the the danger rating levels relate to the severity of the conditions. In the end, the present format of our question with two sliders for each danger rating level seems to provide the most flexible and tangible way for participants to express their perception.

In future studies, the patterns we identified could be presented to participants graphically to pick the one that aligns with their personal understanding of the danger rating scale the best. We think that this would be a meaningful way for exploring this topic further.

**To address this reviewer concern, we will a) discuss the potential challenges of our question format in more detail in the limitation section, and b) we will discuss our perspective on how to best move forward with this research in the conclusion section in more detail.**

Please note that our sample includes 482 individuals (15.1% of the sample) who completed an avalanche safety course aimed at aspiring avalanche professionals. As described in the result section, these

participants were not more prevalent in the group that indicated that the step size between the danger rating levels increases as we go up in the scale. We believe that the most likely reason for this is that these more highly trained individuals perceive the danger scale the same way as the rest of the sample. This interpretation has been confirmed in various casual conversations with Canadian avalanche professionals. However, we cannot conclusively exclude that there might be challenges with the way we asked this question.

**Similar to above, we will address this concern by a) discussing the potential implications of these challenges in more detail in the limitation section of our manuscript and b) proposing to include avalanche professionals and avalanche forecasters more explicitly in future studies.**

**1.5 Design of avalanche danger scale use question**

**Reviewer Comment:**

*As mentioned by the authors, the "use" question was relatively crude. It only asks to what degree the decision to go into the backcountry depends on the danger level, and not if the danger level affects terrain choices in the backcountry. It may also be mentioned that category 1 encompasses category 4 (both imply that the respondents rely mainly on the danger scale).*

**Author Response:**

Studying the effect of the danger rating level on in-field risk management processes is much more complex and clearly beyond what currently can be accomplished with an online survey.

While we agree with the reviewer that category 1 and 4 both highlight a strong reliance on the danger rating, relying mainly on the danger rating to go into the backcountry is very different from mainly using it to decide not to expose oneself. Hence, the question provides insight into both the reliance on the danger rating and the acceptable threshold for going into the backcountry.

**No action is required for this comment.**

**1.6 Choice of latent classes**

**Reviewer Comment:**

*The authors use a latent class mixed effects model. This seems like an adequate approach, as it can identify interesting response patterns in the data. However, as noted by the authors, the model did not converge in the sense that the model fit kept improving at a higher number of latent classes. Instead, the authors stopped the process when the new latent classes did not provide meaningful information. I do not doubt that the research group took great care in analyzing the data and determining the number of latent classes. However, I still fear that they read more into the data than what is actually there. It seems reasonable to divide the responses into convex, linear, and concave functions, with narrow and wide spreads. However, although the classes are significantly different from each other, they are all relatively linear from a practical perspective (eyeballing the graphs, the step size in mid-points between different levels appears to be about 15-20 for all functions). Based on the AIC and BIC, all seven classes should be included in the model, but so should the discarded classes. This makes me wonder how much we gain from including so many classes and how much we should read into the differences.*

**Recommendation:**

*Perception of the avalanche danger scale:*

- *It would help if we knew why the kept classes are more meaningful than the discarded classes. I therefore recommend that a set of discarded classes are presented in an online appendix. These classes should consist of the first e.g., 3-6 classes that improves fit but that the authors deemed meaningless. I would also like to see a short discussion of why the discarded classes have less meaning than the included classes.*
- *Since the differences in functional form are relatively small, it would help if the authors discussed the practical implications of the differences. It would also be interesting to know which analytical functional form the different graphs represent (i.e., how far from an exponential scale are they?).*
- *If the practical implications of the differences are important, I think that you should put more emphasis on the fact that there was no clear pattern among people who perceived the danger scale as convex with narrow spreads.*

**Author Response:**

As described in the manuscript, our selection of the latent class solution was based on both statistical evidence and interpretability (L356-362). While the AIC and BIC indicated that adding more and more classes resulted in a solution that represented the data better and better, the identified differences became less meaningful and practically irrelevant even though they were still statistically significant.

During our analysis, we examined the different class solutions in detail and studied the pathways between the different classes from one solution to the next. **To address this reviewer concern, we will add supplementary material that includes visualizations of all the different class solutions (similar to Fig. 5) and discusses how individuals move between the classes of the different solutions. We believe that this will provide the necessary insight to better understand our choice of the 7-class solution and the differences between the classes.** See our draft of the supplementary material at this end of this document. The practical implications are discussed in the discussion section.

Given that only one of the classes exhibits a concave functional form (common use of the term), we do not think it is meaningful to explicitly test how close the observed patterns are to the theoretical exponential form of the danger scale.

**1.7 Labelling of latent classes**

**Reviewer Comment:**

*Conditional inference trees offer a novel and interesting approach to understand the association between background variables and response patterns. In this paper, the trees are used to understand groupings according to 1) the share of correct responses on danger level rank question, 2) latent class combinations with respect to the functional form of the danger scale, and 3) latent class combinations with respect to use of the danger scale.*

*The conditional inference tree is very helpful to understand the association between background variables and the ability to correctly rank the different danger levels. The authors make an admirable effort to explain the results. However, the final nodes of the inference trees for perception of avalanche hazard and use are very complex and difficult to understand. The reason is that the nodes represent*

*different combinations of latent classes. In the case of use of the danger scale, the latent classes further represent different combinations of user types. The problem is made worse because the different latent classes are ordered in a way that doesn't help the reader. I understand that the statistical software labels the classes and that the authors use these class names to enable replication. However, it makes it cumbersome to read the text. I had to scroll up and down to recall what the different classes represent, and make notes, to have a chance to interpret the figures.*

***Recommendation:*** *I think that it would be beneficial if the authors renamed the classes so that the follow a logical order and given names that describes the class (e.g., for perception: class 1 – convex, class 2 - linear, class 3 – concave. For user group; class 1 – rely mainly on danger level at low levels, class 2 – rely mainly on danger level at high levels, class 3 – rely mainly on other information in avalanche bulletin).*

**Author Response:**

This relates to Reviewer Comment 2.16, which suggests a more intuitive color scale for the classes, as well as Comments 2.2 and 2.4 that discuss the structure of the danger scale perception question results section.

We appreciate this comment and agree that our choice of labels was not very reader friendly. As pointed out by the reviewer, the reason for the numeric labels was to maintain the link with the classes in R.

**To address this reviewer concern, we will create more intuitive labels for the different classes and use them consistently in the text and figures as suggested. We will also try to find a better color scheme to make it easier to see the higher-level patterns in the prevalence of the clusters (e.g., convex, linear, concave). See our current draft of the supplementary material at the end of this document for our current thinking on the class labels.**

**1.8 Recall question**

**Reviewer Comment:**

*I am not sure why it is interesting to analyze if people recall the names of the different danger levels. If kept in the paper, I would like to see a clearer discussion about what we learn from the results.*

**Author Response:**

There are two main reasons for including the recall question in our survey and analysis. First, we grounded the design of our survey in Bloom's taxonomy of learning objectives, whose levels are remember, understand, apply, analyze, evaluate, and create. In our context, only the first three levels are relevant, which we implemented in our study with the recall, perception and use questions. Hence, we believe that the recall question can provide us with important insight about the base familiarity with the scale.

On a more practical level, the danger rating level terms are the primary means for communicating the severity of the avalanche hazard conditions, and not knowing the full scale and proper order of the terms would prevent somebody from meaningfully interpret a rating. This is particularly important in North America because, in contrast to European warning services, North American warning services only use these terms to describe avalanche conditions and not numbers (e.g., 2 – Moderate, 3 –

Considerable). In addition, there has been discussions about the usefulness and intuitiveness of the term "Considerable" for decades. Hence, examining whether bulletin users know the terms and can put them in the right order seems useful to us.

**To address this reviewer concern, we will discuss the usefulness of the question and relevance of the results in more detail in the methods and discussion sections.**

**1.9 Structure of Section 3.3**

**Reviewer Comment:**

*Section 3.3. I think that it would help to first describe the different classes (give some intuition for them) and then describe how many participants fall into each class. I suggest to organize the classes into Convex, Linear, and Concave.*

**Author Response:**

This comment relates to Reviewer Comment 1.7 and 2.16, which suggest a more intuitive labelling and colouring of the classes, as well as Comments 2.2 and 2.4 that discuss the structure of the danger scale perception question results section.

We find it important to explicitly describe the exclusion criteria and our selection process for the class solution explicitly before getting into the details of the results. **However, to address this reviewer suggestion, we will change the labels of the classes (as described in our response to Comment 1.7) and revise the description of the classes to make them more intuitive and easier to follow.**

**1.10 Figure 5**

**Reviewer Comment:**

*Figure 5. I would like to see a caption and figure order that is consistent and easy to remember. I found the figure very confusing, because the caption mixed "a" to "f" with the class labels, and the class labels did not follow any logical order (to me as a reader).*

**Author Response:**

We can see that the current presentation of the different classes was not as intuitive as we thought. The individual classes/charts are currently ordered roughly from the most concave to the most convex (geometric interpretation of the term) and from the narrowest to the widest ranges. The present order emerged from how the classes evolve from the different class solutions. See Figure S-2 in the supplementary material at the end of this document for an overview of how the classes and their order emerged from the analysis. Numbers refer to the classes as they are labelled in R and in the text.

**To address this reviewer comment, we will improve the labelling of the classes in both Fig. 5 and in the text (see our response to Comment 1.7 as well) and describe the ordering of the classes in the caption of Fig. 5 as well. See our current draft of the supplementary material at the end of this document for our current thinking.**

**1.11 Structure of Section 3.4**

**Reviewer Comment:**

*Section 3.4. I would have preferred to read a description of the order of the different classes before the presentation of the list, as this would have helped me understand the order. If possible, order the classes based on how "correct" the strategy is. If that is not possible, give the classes a name that describes the group. This will help when the reader is trying to understand figure 8.*

**Author Response:**

**To address this comment, we will a) move the description of the class order ahead of the bullet list that describes the individual classes, and b) give each class a more meaningful label that can be used consistently in the text and figures.**

Please note that there is no correct strategy for using the avalanche bulletin information as it depends on the needs and desires of the user.

**1.12 Discussion: Relationship between functional form and danger rating use.**

**Reviewer Comment:**

*Discussion: I would like to see a discussion of why we should expect to see a relationship between a specific functional form of the danger scale and a certain use of the scale. It is not clear to me if one of the use strategies is more correct than another and if a e.g., convex functional form would imply more or less reliance on the scale relative to other information in the bulletin.*

**Author Response:**

The responses to the danger scale use question provide insight into both participants' reliance on the danger scale and their personal danger rating level thresholds that prevent them from going into the backcountry. Our hypothesis was that differences in the perception of the severity of the conditions would potentially lead to different thresholds. In other words, we expected participants who perceive avalanche conditions to be less severe under an Extreme rating, for example, to be more likely go into the backcountry under this rating than people who perceive the conditions to be more severe. More specifically, it seems plausible that participants who associated a wide range of possible conditions with Extreme (Fig. 5, Panel g, Class 3) might feel more comfortable going into the backcountry under this danger rating level or rely on other bulletin information to make that decision than participants who associate Extreme with a very high and narrow range of severities (Fig. 5, Panel e, Class 1). However, our analysis did not find evidence of this relationship, which can at least partially be explained by the only subtle differences in the perception clusters.

**To address this concern, we will explain the reasoning behind our hypothesis in more detail in the methods section and refer back to it in the discussion section. Also see our response to Reviewer Comment 1.14.**

**1.13 Discussion: Comment on validation of bulletin user scale**

**Reviewer Comment:**

*Discussion: the authors say that the results indirectly validate the Bulletin user typology. I would like to argue that this is incorrect. Both the user typology and the question about use of the avalanche scale*

*asks how a person uses the avalanche bulletin, i.e., they are two ways to ask the same thing. The results therefore suggest that we can measure self-assessed use with two methods. To validate if the bulletin user typology represents actual differences in use, we would have to look at actual use. I would like to see a more nuanced discussion about this.*

**Author Response:**

We agree with the reviewer that a proper validation of the bulletin user typology would require its own study and a different research approach. **Hence, we will delete this comment from the manuscript.** Also see our response to Reviewer Comment 2.5.

**1.14 Discussion: Detailed discussion of risk perception background**

**Reviewer Comment:**

*Discussion: The discussion about risk perception seems misplaced. The questions in the survey ask about the objective risk for widespread very large avalanches and not about ability to mitigate the hazard. Although the discussion is fascinating, I don't think that it belongs in this paper.*

**Author Response:**

We hope that our response to Reviewer Comment 1.12 explains why we expected a relationship between danger rating level perception and use and provides context for our risk perception discussion (L600-622).

**To address this comment, we will a) move the initial part of the discussion to the methods section to better explain why we expected this relationship, and b) shorten the section a bit. This also helps address Reviewer #2's general sentiment that the manuscript is too long.**

**1.15 Availability of questionnaire and additional survey information**

**Reviewer Comment:**

*All datafiles are available on osf. This is great! However, I would also like to see that the questionnaire (in full) was publicly available. Information about the length (average completion time) should be provided in the text, together with information about where in the survey the questions were placed. The latter is important since the task may have been perceived as relatively challenging for the respondents.*

**Author Response:**

Please note that the full questionnaire is publicly available in the master's thesis of Finn (2020) as mentioned on L 106.

**However, to address this concern and make the full questionnaire more easily accessible, we will also add a PDF with the full questionnaire to the online repository. Furthermore, we will add a brief statement in Section 2.1 (Survey Design) describing where the danger scale questions examined in this study were located in the survey.**

**Responses to Reviewer #2**

**GENERAL COMMENT**

We thank Frank Techel for his constructive review and helpful comments. We appreciate the encouraging comments about the quality of our study and its contribution to the scientific literature. Please see below for our detailed responses to specific his comments and suggestions.

**RESPONSES TO SPECIFIC COMMENTS**

**2.1 Length of manuscript**

**Reviewer Comment:**

*The manuscript is rather long. Shortening some sections might help the reader to focus on the key findings.*

**Author Response:**

We agree with the Frank's comment that our manuscript is long, and that a more concise and focused paper would be better.

**To address this comment, we will tighten the text throughout the manuscript and ensure that figures and tables are used strategically to present the important information as efficiently as possible**. See our responses to other comments (e.g., 2.2 to 2.4, 2.15, 2.18) for more specific information.

**2.2 Information presented in results section**

**Reviewer Comment:**

*Each of the Result sections 3.1, 3.2 and 3.4 starts with the number of responses available for the respective analysis, and introduces the number of significant classes obtained from the LCME model (e.g. 304-308, 350-359, 416-420). While this makes it very transparent, which data was excluded, I wonder whether there would be a way to present this information in a more concise way to make these sections more focused. For instance, could this information be moved to either a small table and/or a short section at the end of the Methods-Section? By doing so, the reader would still be able to find this information, if interested, but could focus more easily on the findings.*

**Author Response:**

This comment relates to Reviewer Comments 1.9 and 2.4, which also refer to the structure and presentation of the results sections.

It is important to us to be completely transparent about the inclusion/exclusion criteria and our choices during the analysis process. While some of the information could potentially be moved into the methods section, we believe that the results section is the proper location to describe these aspects as they emerge during the analysis. We do not want to move this information into an appendix as it will likely get lost.

**However, to address this comment, we will look at this information carefully and shorten it as much as possible.**

**2.3 Presentation of results of recall analysis**

**Reviewer Comment:**

*Consider moving some of the results to a table. This would provide an easy-to-read overview of some of the results and may also allow to shorten the text. For instance, moving the findings on l304-312 to a table.*

**Author Response:**

**To address this comment, we will transfer some of the results of the recall question into a table.** While this will shorten the text by a few lines, it will likely not save any space overall.

**2.4 Presentation of results of perception analysis**

**Reviewer Comment:**

*A large part of this section describes in detail what is shown in Figure 5. While this is certainly a very interesting way of analyzing the survey respondents' perception of the danger levels, I feel that this entire section could be shortened. Firstly, the slopes shown in the plots of the three largest classes (classes 2, 4 and 1 combined 85%) look rather similar even though statistically different. I understand that the authors also consider these variations as rather subtle (e.g., l508, l597). Secondly, these three classes combined seem equally frequent in the three nodes in the CTree analysis shown in Figure 6 (80 to 85%?), suggesting that the variables used to explain differences in how participants perceive the danger scale fail to really differentiate between class membership. My interpretation of these findings is that most respondents understand the scale as a linear one (about 60%), with some respondents having a (slight?) tendency towards a concave or a convex interpretation, and that the variables describing the respondents' skills and experience can't really explain the LCME class membership. The take-home message is that respondents perceive the danger scale primarily as a linear scale, and thus different compared to the scientific interpretation. This is a robust and relevant finding. – I propose to emphasize the key findings, maybe at the end of this section, and to consider shortening this section.*

**Author Response:**

This comment relates to Reviewer Comments 1.9 and 2.2, which also refer to the structure and presentation of this section.

We agree with Frank that this section could be shortened, and better class labels will make it easier to read (see response to Reviewer Comment 1.7). While Frank accurately summarizes the main take-home points of our analysis, such a synthesis of the results is typically presented in the discussion section. We provide the most concise summary of our results in the opening paragraph of the result section.

In our opinion, the purpose of the result section is to comprehensively present the relevant details of the analysis to allow readers develop their own opinions about the main take-home points of the study. The content and detail of the description in the results section is primarily driven by the analysis results. Given that there is considerable judgment involved in our analysis (e.g., selection of parameters, choice of latent class solution), we think this level of detail is important. In fact, Reviewer #1 has requested

additional details about the perception analysis (Comment 1.6). Readers not interested in these details are welcome to skip the results section and go directly to the discussion.

**To address this concern, we will tighten the text of the results sections as much as possible and rely more heavily on the figures to present the details of the analysis.**

**2.5 Summary of results using bulletin user typology**

**Reviewer Comment:**

*I feel that it could be beneficial to the reader, and particularly to those who are interested in gaining insights on the different bulletin user groups, to summarize the key results in a short section and a graph/table, potentially like the following figure:*

| Bulletin user typology | Recall and order of DL | Perception of DL | Use of DL in trip planning | Background variable (median, mode) | N (%) |
|---|---|---|---|---|---|
| A | x% five levels x% levels in correct order | --- | --- | years of experience, number of days, … | 45 (1.3%) |
| B | x% five levels x% levels in correct order | xx% linear x% concave | Barplot similar to subplots in Fig 5 | years of experience, number of days, … | … |
| C | … | … | Barplot similar to subplots in Fig 5 | … | … |
| D | … | … | … | … | … |
| E | … | … | … | … | 45.4% |
| F | … | … | … | … | … |

*Personally, I would have liked to see such an analysis/summary, as it would have allowed me to link the key findings more easily to different bulletin user groups. Sections 3.1, 3.3 and 3.4, and the corresponding explanations in the manuscript provide similar information, though always from the perspective of the CLME class assignment. However, I had trouble linking the class assignment to the user typology shown in Table 1, which I find a helpful and intuitive classification. Therefore, I propose to look at the data from this perspective too. Such a section may also support the statement on l673-675. Furthermore, linking bulletin user class and the most relevant background variables could provide a helpful overview describing the survey respondents (described also on l 185-199).*

**Author Response:**

We appreciate this suggestion, and we have thought about our response long and hard. After careful consideration, we would prefer not to expand the manuscript with presenting all results segmented according to avalanche bulletin user types. There are three main reasons for our position:

First, the objective of this study was to provide a first overview of how avalanche bulletin users remember, perceive and use the avalanche danger scale during trip planning. We used self-reported avalanche bulletin user type to a) guide participants through the survey and b) interpret the response patterns in the danger rating use question. However, a characterization of the bulletin user types was not an explicit objective of our study. So, while Frank's suggestion is intriguing, it would add an new research objective to our study that is considerably different from the existing ones.

Second, a proper and insightful characterization of avalanche bulletin user types will require a dedicated study with a sample that represents all bulletin user types more evenly. Comparing 45 Type As against 1,451 Type Es does not seem very meaningful. Our research group is currently working with a number of avalanche warning services (e.g., SLF, Euregio, Colorado Avalanche Information Center, Avalanche Canada) to develop research panels that will facilitate such studies in the not so distant future.

Third, our manuscript is already too long and adding an additional research objective would add a considerable amount of additional text.

**Because of all these reasons, we would prefer not to add a new section describing our survey results split up by avalanche bulletin user types. However, we will go through the entire manuscript and carefully consider any references to the bulletin user typology to make sure the focus of our study is clear.** Also see our response to Reviewer Comment 1.13.

**2.6 Practical implications for risk communication**

**Reviewer Comment:**

*A reduction of five to four levels is discussed thoroughly in this section. Beside the numerous arguments for and against such a change (l679-803), it might be worth taking up the point that avalanche forecasts in Europe, and thus the avalanche danger scale, not only target recreational users but are also an important source of information for decision-makers responsible for the safety of the public in residential areas or on transportation networks. For these decision-makers, but also the public at large, the two highest danger levels are particularly relevant to communicate these rare, but very dangerous situations clearly.*

*Furthermore, a different approach is used by the Swiss avalanche warning service. As of this winter, in the published forecasts the danger level will be shown together with a qualifier (-, =, +) indicating where within the level the avalanche hazard is expected (e.g., SLF, 2022a, SLF, 2022b; described for instance in Techel et al, 2022). This development would also go against the discussed reduction to four levels and could maybe be mentioned/discussed?*

**Author Response:**

We thank Frank for these additional suggestions. Please note that we mention in the introduction (L53-56) that the avalanche danger scale has different applications in Europe and North America. While the primary focus of the North American Public Avalanche Danger Scale is public risk communication, the European system is used in a wider range of applications that also includes providing warnings for residential areas and transportation networks.

**To address this comment, we will remind readers of the different applications of the avalanche danger rating scale in Europe and North America in the discussion section. In addition, we will add a brief comment about the recent introduction of danger rating sublevels in Switzerland.**

This also relates to Reviewer Comment 2.25.

**2.7 Addition to abstract**

**Reviewer Comment:**

*L16-17: This statement is correct though one could add that this linear perception is also in line with the danger scale being an ordinal scale.*

**Author Response:**

Thanks for highlighting the section in the abstract where the nature of the linear interpretation is not described very clearly. **We will improve this description in the revised manuscript to ensure that it is clearly distinct from the characteristics of an exponential scale.** However, being ordinal does not say anything about the functional form of the scale.

**2.8 Local versus regional**

**Reviewer Comment:**

*L38, 102 (and maybe other places): I am not sure if calling the Canadian avalanche warning services local is the most appropriate term. I would find regional forecasting more suitable. For instance, the European Avalanche Warning Services provide a forecast for a region.*

**Author Response:**

Thanks for pointing out this interpretation of our writing. **We will simply delete 'local' in reference to avalanche warning services as it is not important.**

**2.9 Danger level versus danger rating level**

**Reviewer Comment:**

*A terminology question: do you call the danger levels (Figure 1) in North America also danger rating levels (as for instance in title of Section 3.4)? - Consider using danger level throughout, if appropriate.*

**Author Response:**

This is how we see the distinction between these two terms:

- *Danger rating levels* refer to a particular rating on the scale. Our research focused on participants recall, perception and use of the different danger rating levels.
- *Danger levels* refer to the severity of the actual hazard conditions. This is related but different from the danger rating levels, which are human descriptions of what is going on.

To make it easier to distinguish between these two terms, we use "danger rating levels" when referring to the scale levels and "severity of conditions" when talking about the actual hazard level.

**We will carefully go through the manuscript to ensure we explain the meaning of these terms clearly and use of them consistently.**

**2.10 Survey design**

**Reviewer Comment:**

*Section 2.1 Survey Design: The survey design is described clearly and with sufficient detail. As I am not an expert in designing surveys, I can't judge, for instance, what effect the provision of numerical sliders may have on the responses.*

**Author Response:**

**No action required for the comment.**

**2.11 Typo**

**Reviewer Comment:**

*L170: spend → spent*

**Author Response:**

Thanks for highlighting this typo. **We fixed this sentence.**

**2.12 Inconsistency in number of responses**

**Reviewer Comment:**

*L185 vs. 207: Maybe check whether there is a typo in one of the numbers: 3195 responses (L185) minus the 42 A-responses (l343) would result in 3153 rather than 3130 (l207).*

**Author Response:**

Thanks for highlighting this inconsistency. **We will carefully examine the numbers and either update them or provide an explanation for the discrepancy.**

**2.13 Typo**

**Reviewer Comment:**

*L285: estimate → estimated*

**Author Response:**

Thanks for highlighting this typo. **We fixed this sentence.**

**2.14 Missing word**

**Reviewer Comment:**

*L322-324: is there a word missing in this sentence? After "training"?*

**Author Response:**

Thanks for highlighting this oversight. **We fixed this sentence.**

**2.15 Suggestion for shortening text**

**Reviewer Comment:**

*L325-330: In case you intend to shorten some sections, maybe the text describing the two final splits in Figure 4 could be omitted.*

**Author Response:**

**We will consider this when tightening the text of the complete manuscript.**

**2.16 Colors and labels for latent classes**

**Reviewer Comment:**

*Sect. 3.3 and Sect. 3.4: the classes obtained with LCME model are referred to by numbers (1-7, 1-6) using the same color scheme in Fig. 6 and 8. While the legend titles in these figures indicate that classes are different, this fact could maybe be supported by using different color schemes, and maybe by using once numbers and once letters for labeling.*

**Author Response:**

This relates to Reviewer Comment 1.7, which suggests a more intuitive labelling of the classes, as well as Comments 2.2 and 2.4 that discuss the structure of the danger scale perception question results section.

**We will completely revise the labeling of the classes and choose a more intuitive color scheme for making the figures easier to interpret.**

**2.17 Adding information on bulletin user types**

**Reviewer Comment:**

*L425-437: Consider including the proportion of self-identified bulletin user types – maybe the proportion of B and C combined vs. the proportion of E and F combined – to the description of the six classes.*

**Author Response:**

This relates to Reviewer Comment 2.5, which proposes a more avalanche bulletin user type focused presentation on the results. It also relates to Reviewer Comment 2.23.

Consistent with our earlier response, we would prefer to de-emphasize the link to the bulletin user typology in this manuscript. See our response to Comment 2.5 for details. **We will carefully go through the manuscript and assess where adding summary statistics of bulletin user types is useful without detracting from the main objectives of our study.**

**2.18 Presentation of results**

**Reviewer Comment:**

*L444-448: This is, of course, just a personal preference, but maybe consider moving the overall results to the beginning of this section, followed by the detailed analysis.*

**Author Response:**

**Based on all the feedback received, we will carefully revise the results sections to make them more focused and concise.**

**2.19 Description of danger scale**

**Reviewer Comment:**

*L499: It might be worth repeating that the avalanche danger scale is primarily an ordinal scale (l47), with categorical descriptions of the danger levels. A large share of the respondents got the order of the levels right.*

**Author Response:**

Thank you for this suggestion. **We will add this reminder about the nature of the danger scale in the text.**

**2.20 Reference to Techel et al (2022)**

**Reviewer Comment:**

*L510-513: Maybe of interest: a recent study exploring numerous observations related to the contributing factors of avalanche hazard and the corresponding increase in the severity of the hazard with increasing danger level is Techel et al. (2022). This study also shows changes within the forecast danger levels.*

**Author Response:**

Thanks for this suggestion. **We will add a reference to Techel et al. (2022) in this section of the discussion.**

**2.21 Presentation of danger scale by European warning services**

**Reviewer Comment:**

*521-523: Just an observation: At least some warning services in the European Alps present the danger scale as a scale which shows an exponential increase. Examples include the websites from the Swiss avalanche warning service (SLF, 2022a), as well as the avalanche warning service in Tyrol-South Tyrol-Trentino (avalanche.report, 2022). However, there are also European warning services where no such presentations can be found. The question you raise, would therefore indeed be an interesting one to answer: do avalanche forecasters and educators themselves perceive this non-linear increase in the severity of avalanche hazard with increasing danger level, and if they do, do they consider it important to communicate? Given the findings on the use of the danger scale, with danger levels having a considerable impact on stated decision-making during trip planning, how important is it that users have a different perception of the danger scale?*

**Author Response:**

Thanks for explaining the European situation to us in more detail. **We will adjust the text accordingly to more clearly highlight that the presentation of the format is not consistent in Europe.**

Your final question is an interesting one. Given that the perception of the scale (at least how we measured it) does not seem to have an influence on participants' use of the scale, does it really matter

whether they perceive it as a linear or exponential scale? Our opinion is that it is probably an uphill battle to continuously educate the public that the danger scale is an exponential scale. Instead of trying to educate bulletin users scientifically, we think that including practical behavioral guidance (e.g., terrain suggestions) is probably a more promising path for helping bulletin users manage their personal avalanche risk more meaningfully.

**To address this comment, we will better highlight our position on this question more clearly in the final paragraph of Section 4.3.**

**2.22 Additional reason for linear perception**

**Reviewer Comment:**

*L525: Another reason for this linear perception is maybe also the fact that the scale is primarily an ordinal scale with categorical descriptions.*

**Author Response:**

We completely agree. A linear interpretation seems to most likely default for an ordinal scale. **We will add this point to the text.**

**2.23 Relation to bulletin user types**

**Reviewer Comment:**

*L591-593: As suggested before, it would be nice if you could link this statement to a figure or short section emphasizing the relationship between the bulletin user typology and the results.*

**Author Response:**

This relates to Reviewer Comments 2.5 and 2.17.

**As discussed in our response to Comment 2.5, we would prefer to de-emphasize the link to the bulletin user typology in this manuscript. See our response to comment 2.5 for details.**

**2.24 Reference to existing of terrain-based tools: skitourenguru.ch**

**Reviewer Comment:**

*L714-717: maybe worth mentioning that such terrain-based tools are already operational, as for instance the website skitourenguru.ch, where back-country ski touring routes in the European Alps are risk-rated according to the forecast avalanche conditions and the terrain*

**Author Response:**

**We will consider adding a reference to the skitourenguru.ch website.** We have some personal reservations about this website, and our vision for terrain guidance tools is different, but this is a discussion that is beyond the scope of this paper.

**2.25 Clarification about bulletin users**

**Reviewer Comment:**

*L764-765: Maybe add "in North America" after "bulletin products", as this statement would not be true in Europe.*

**Author Response:**

Thanks for highlighting this inaccuracy. **We revised the sentence as follows to be more accurate:**

*Our research fills an important gap in understanding how recreationists, the primary target audience of bulletin products in North America and an important bulletin user in Europe, interact with danger ratings.*

**2.26 Reference to Terum et al. (2022)**

**Reviewer Comment:**

*L773-775: maybe of interest as you are mentioning how trends in forecast danger level are perceived, Terum et al. (2022) address this topic in their study*

**Author Response:**

Thanks for making us aware of this recent publication. We will include it as a new reference.

**Supplemental Material**

**A user perspective on the avalanche danger scale – Insights from North America**

Abby Morgan[1], Pascal Haegeli[1], Henry Finn[1, 2], Patrick Mair[3]

[1]School of Resource and Environmental Management, Simon Fraser University, Burnaby, V5A 1S6, Canada
[2]School of Social and Political Science, University of Edinburgh, Edinburgh, EH8 9LD, UK
[3]Department of Psychology, Harvard University, Cambridge, MA 02138, United States

*Correspondence to*: Pascal Haegeli (pascal_haegeli@sfu.ca)

This document includes additional figures in support of the latent class analysis of the danger rating perception question. This includes:

- AIC and BIC for latent class solutions with 2-9 classes (Figure S-1)
- Overview flow chart highlighting how classes evolve from one class solution to the next (Figure S-2)
- Visualization of each class solution (Figures S-3 to S-11).

[Figure]

**Figure S-1: AIC and BIC for latent class solutions with 2-9 classes.**

[Figure]

**Figure S-2: Overview flow chart highlighting how classes evolved from one class solution to the next. Labels of classes include class identifiers, general description of perception pattern, and size of class. Classes are grouped and coloured based on their second polynomial parameter estimate: concave (> 0.75; blue), linear (> -0.75 and < 0.75; yellow), and convex (< -0.75; red). Terms for the width of the ranges are qualitative and relative within a single model. Lines indicate the size of participant pathways between classes with the line width being proportional to the number pf participants. Lines are only shown if the pathway represents at least 10% of the class in the higher-class solution.**

[Figure]

**Figure S-3: Visualization of 1-class solution. Panel presents the danger rating regression line (black line) and severity ranges for each level (thick vertical lines) from the regression analysis and the severity range distributions of participants' answers associated with this class. Overall sample size is shown above the top left corner of the charts, and class id above the top right corner.**

[Figure]

**Figure S-4: Visualization of 2-class solution. Each panel presents the danger rating regression line (black line) and severity ranges for each level (thick vertical lines) from the regression analysis and the severity range distributions of participants' answers associated with the particular class. Classes are arranged by identified perception pattern (generally from concave to linear and convex) See Figure S-2 for related information. Class sizes are shown above the top left corner of the charts, and class ids above**

35 **the top right corner.**

[Figure]

**Figure S-5: Visualization of 3-class solution.** Each panel presents the danger rating regression line (black line) and severity ranges for each level (thick vertical lines) from the regression analysis and the severity range distributions of participants' answers associated with the particular class. Classes are arranged by identified perception pattern (generally from concave to linear and convex) See Figure S-2 for related information. Class sizes are shown above the top left corner of the charts, and class ids above the top right corner.

**Figure S-6: Visualization of 4-class solution.** Each panel presents the danger rating regression line (black line) and severity ranges for each level (thick vertical lines) from the regression analysis and the severity range distributions of participants' answers associated with the particular class. Classes are arranged by identified perception pattern (generally from concave to linear and convex) See Figure S-2 for related information. Class sizes are shown above the top left corner of the charts, and class ids above the top right corner.

[Figure]

50    **Figure S-7: Visualization of 5-class solution. Each panel presents the danger rating regression line (black line) and severity ranges for each level (thick vertical lines) from the regression analysis and the severity range distributions of participants' answers associated with the particular class. Classes are arranged by identified perception pattern (generally from concave to linear and convex) See Figure S-2 for related information. Class sizes are shown above the top left corner of the charts, and class ids above the top right corner.**

55

[Figure]

**Figure S-8: Visualization of 6-class solution. Each panel presents the danger rating regression line (black line) and severity ranges for each level (thick vertical lines) from the regression analysis and the severity range distributions of participants' answers associated with the particular class. Classes are arranged by identified perception pattern (generally from concave to linear and convex) See Figure S-2 for related information. Class sizes are shown above the top left corner of the charts, and class ids above the top right corner.**

60

[Figure]

**Figure S-9: Visualization of 7-class solution.** Each panel presents the danger rating regression line (black line) and severity ranges for each level (thick vertical lines) from the regression analysis and the severity range distributions of participants' answers associated with the particular class. Classes are arranged by identified perception pattern (generally from concave to linear and convex) See Figure S-2 for related information. Class sizes are shown above the top left corner of the charts, and class ids above the top right corner.

[Figure]

**Figure S-10: Visualization of 8-class solution. Each panel presents the danger rating regression line (black line) and severity ranges for each level (thick vertical lines) from the regression analysis and the severity range distributions of participants' answers associated with the particular class. Classes are arranged by identified perception pattern (generally from concave to linear and convex) See Figure S-2 for related information. Class sizes are shown above the top left corner of the charts, and class ids above the top right corner.**

[Figure]

**Figure S-11: Visualization of 9-class solution. Each panel presents the danger rating regression line (black line) and severity ranges for each level (thick vertical lines) from the regression analysis and the severity range distributions of participants' answers associated with the particular class. Classes are arranged by identified perception pattern (generally from concave to linear and convex) See Figure S-2 for related information. Class sizes are shown above the top left corner of the charts, and class ids above the top right corner.**